# Skin basal cell carcinomas assemble a pro-tumorigenic spatially organized and self-propagating Trem2+ myeloid niche

Daniel Haensel [1,2], Bence Daniel [3,4], Sadhana Gaddam[1,2], Cory Pan [1,2], Tania Fabo [1], Jeremy Bjelajac [1], Anna R. Jussila [1,2], Fernanda Gonzalez[1,2], Nancy Yanzhe Li [1,2], Yun Chen[5,6], JinChao Hou[5], Tiffany Patel[1,2], Sumaira Aasi[2], Ansuman T. Satpathy [3,4,7] & Anthony E. Oro [1,2] ✉

Cancer immunotherapies have revolutionized treatment but have shown limited success as single-agent therapies highlighting the need to understand the origin, assembly, and dynamics of heterogeneous tumor immune niches. Here, we use single-cell and imaging-based spatial analysis to elucidate three microenvironmental neighborhoods surrounding the heterogeneous basal cell carcinoma tumor epithelia. Within the highly proliferative neighborhood, we find that TREM2+ skin cancer-associated macrophages (SCAMs) support the proliferation of a distinct tumor epithelial population through an immunosuppression-independent manner via oncostatin-M/JAK-STAT3 signaling. SCAMs represent a unique tumor-specific TREM2+ population defined by VCAM1 surface expression that is not found in normal homeostatic skin or during wound healing. Furthermore, SCAMs actively proliferate and self-propagate through multiple serial tumor passages, indicating long-term potential. The tumor rapidly drives SCAM differentiation, with intratumoral injections sufficient to instruct naive bone marrow-derived monocytes to polarize within days. This work provides mechanistic insights into direct tumor-immune niche dynamics independent of immunosuppression, providing the basis for potential combination tumor therapies.

While the advent of immune checkpoint blockade (ICB) therapies has revolutionized approaches to cancer treatment, the efficacy of these treatments remains inconsistent[1,2]. Newer strategies aimed at improving ICB efficacy with additional blocking antibodies to components of the tumor microenvironment (TME) that enhance antigen presentation to T cells have had only incremental success[2–4]. Alongside these efforts, single-cell genomic technologies have revealed that resistance to ICB includes extensive tumor heterogeneity and

plasticity, with drug interventions inducing tumor evolution programs that circumvent therapy[5,6]. A better understanding of the immunoregulatory concepts that explain the origin, assembly, and dynamics of the heterogeneous tumor epithelium immune microenvironment is needed.

Skin basal cell carcinomas (BCCs) are ideal for studying the interplay between targeted therapies and epithelial heterogeneity and plasticity because of the high incidence and the ability to serially

[1]Program in Epithelial Biology, Stanford University School of Medicine, Stanford, CA, USA. [2]Department of Dermatology, Stanford University School of Medicine, Stanford, CA, USA. [3]Department of Pathology, Stanford University School of Medicine, Stanford, CA, USA. [4]Gladstone-UCSF Institute of Genomic Immunology, San Francisco, CA 94158, USA. [5]Department of Pathology and Immunology, Washington University School of Medicine, St Louis, MO, USA. [6]Department of Neurology, Washington University School of Medicine, St Louis, MO, USA. [7]Parker Institute of Cancer Immunotherapy, San Francisco, CA 94305, USA. ✉e-mail: oro@stanford.edu

biopsy from the exact location along a treatment course[7]. Extensive work has focused on tumor epithelial-centric resistance mechanisms to targeted Hedgehog (HH) therapy that include induction of, and selection for, discrete tumor epithelial populations that mediate relapse[7–13]. Recently we identified a persister BCC population marked by the surface marker LY6D that possesses BCC and squamous cell carcinoma-like features[10,13,14]. Local environmental signaling and applied therapies determine the kinetics of LY6D accumulation, providing a defined approach to interrogating the effects of new treatments[14].

While the BCC epithelial states are defined, TME elements that drive and stabilize each tumor state remain poorly characterized. As with other epithelial tumors, ICB via anti-PD1 (aPD1) and anti-PDL1 (aPDL) antibodies have been approved as second-line therapies for BCC patients that have failed targeted therapy. aPD1 therapy shows a response rate of about 30%, suggesting that further understanding of resistance mechanisms is needed[15]. In addition to tumor epithelial heterogeneity, dynamic TME heterogeneity and selection have emerged as a frequent basis for tumor evasion. Deconvolution of the heterogeneous and dynamic immunosuppressive TME components points to stromal populations such as myeloid cells and cancer-associated fibroblasts as immunosuppression and ICB failure drivers[16]. BCC aPD1 clinical trial data implicate myeloid cells, including populations of macrophages, that are enriched in the non-responders who fail ICB[17,18].

Tumor-associated macrophages (TAMs) have been of great interest in ICB due to their ability to enhance immunosuppression through means of suppressing tumor-infiltrating lymphocytes (TILs), such as direct interactions with PD1 via upregulation of PDL1, or secretion of immunosuppressive metabolites or ligands that promote the recruitment of T regulatory cells (Tregs)[19,20]. Subsets of TAMs, such as TREM2$^+$ TAMS, have recently been implicated in promoting an immunosuppressive TME and the simultaneous targeting of these cells in conjunction with aPD1 ICB enhances tumor clearance[21,22]. Furthermore, increased TREM2+ TAMs have been found in patients who fail aPD1 ICB[23,24]. Beyond the identification of particular targetable subsets of TAMs, recent studies have also focused on the origins and maintenance of TAMs, as their removal would eliminate a significant tumor stimulus and remains a therapeutic hurdle[25,26]. Most models of TAM maintenance rely on continual TAM precursor recruitment and subsequent differentiation as the primary source. However, anti-chemokine clinical trials blocking migratory signals like CCL2/CCR2 have fared poorly, suggesting multiple redundant signals may be involved in recruitment or there are other mechanisms promoting the expansion of the TAM population as the tumor grows[27]. These studies reveal the need to understand tumor–immune niche origin, assembly, and dynamics.

Here we use single-cell and spatial expression approaches to identify a distinct TREM2$^+$ VCAM1$^+$ macrophage population within the highly proliferative neighborhood of naïve human BCCs, which we call skin cancer-associated macrophages (SCAMs). SCAMs promote LY6D$^-$ tumor epithelial proliferation via secretion of the ligand oncostatin-M (OSM), a role independent of immunosuppression. We find that SCAMs proliferate within the TME and can be maintained within serially passaged tumors. Tumors recruit and polarize SCAMs from bone marrow-derived monocytes (BMDMs) within days, providing deep insights into the assembly kinetics of the tumor-immune niche that will facilitate rational targeted combination therapies.

## Results

### BCC stromal spatiality indicates distinct global and local neighborhoods

To understand the spatial and regulatory relationships between the tumor epithelium and stroma, we utilized the CO-Detection by indEXing (CODEX) multiplex system, staining for 24 antibodies to interrogate the tumor cellular composition and define global neighborhoods (Fig. 1a and Supplementary Fig. 1a)[28]. Unlike other epithelial tumors, these BCC tumors contained the patients' normal epidermis, which served as a key landmark in our study (Fig. 1b). Individual cells from tumor-associated regions were segmented, excluding the normal epidermis, fluorescent intensities were extracted, and then integrated with Seurat's scRNA-Seq analysis tool (Fig. 1b). In a representative sample, we found 16 different clusters based on varying fluorescent intensities of the 24 antibodies (Fig. 1b and Supplementary Fig. 1b). We identified key cell types including tumor epithelial cells (PANCK$^+$), fibroblasts (CD90$^+$), endothelial cells (CD31$^+$), subsets of immune cells (HLA-DR$^+$ antigen-presenting cells or CD3$^+$ T cells), and their proliferative index (KI67$^+$) with similar proportions of the primary cell types across three patient tumors (Fig. 1c, d).

Cluster analysis of the cell types allowed us to deconvolute the tumor organization into three distinct global neighborhoods with respect to the normal epidermis. Along the z-axis, we noticed that the lower tumor (LT) region had enhanced proliferation marked by KI67 (Fig. 1e and Supplementary Fig. 1d, e). In contrast, tumor epithelial proliferation was reduced compared to the LT near the interface with the patient's epidermis, which we refer to as the upper tumor (UT) (Fig. 1e and Supplementary Fig. 1c–e). Quantitative analysis of the tumor epithelial-containing compartments confirmed the visual observation that the LT had a higher proportion of proliferating tumor cells (Fig. 1h). Finally, in contrast to the tumor epithelium-containing zones, we noted areas interspersed with regions largely devoid of tumor epithelium, dominated by large amounts of CD3$^+$ and HLA-DR$^+$ cells, which we refer to as the immune swarm (IS) (Fig. 1b, e and Supplementary Fig. 1e, f).

At the IS, UT, and LT global neighborhood level, we noted alterations in immune cell subsets, distinct immune cellular behaviors, and spatial localizations (Fig. 1g, I, n). First, we found that the general distribution of HLA-DR$^+$, CD3$^+$FOXP3$^-$ (non-T regulatory cells), and CD3$^+$FOXP3$^+$ (T regulatory cells) cells differed in their compositions within the various global neighborhoods (Fig. 1i, j). Second, we noted a dramatically higher ratio of HLA-DR$^+$ cells to CD3$^+$FOXP3$^-$ cells in the highly proliferative LT compared to the UT (Fig. 1i, j). Beyond the altered ratio of CD3$^+$FOXP3$^-$ cells, we found alterations in CD3$^+$ T cell subtypes, specifically finding that CD3$^+$FOXP3$^+$ T regulatory cells (Tregs) were enriched in the LT (Fig. 1i, j). Third, in the absence of tumor epithelium, we noted an abundance of CD3$^+$ cells, which at times appeared to cluster with one another (Fig. 1i, j). Overall, these global neighborhood observations indicate that the LT is defined by a highly proliferative epithelium, enriched for tumor epithelial-interacting HLA-DR$^+$ cells and CD3$^+$FOXP3$^+$ Tregs.

We previously used single-cell genomics and spatial tools to define HH therapy-resistant (LY6D$^+$) and sensitive (LY6D$^-$) tumor epithelial cell states in BCCs[14]. We found that at a local level, $LY6D^+$ tumor epithelial cells were generally spatially segregated away from the tumor stroma. Globally, RNAScope revealed consistently larger $LY6D^+$ clones within the UT region compared to the LT region (Fig. 1k). To determine how our global CODEX observations relate to our previous findings on tumor epithelial organization, we examined the relation of HLA-DR$^+$ cells to the resistant $LY6D^+$ tumor epithelium (Fig. 1l). As previously shown, the $LY6D^+$ cells are more spatially localized in central portions of tumor nodules and away from the HLA-DR$^+$ cells within the tumor stroma (Fig. 1l).

Overall, our initial analysis identifies three distinct features that define global BCC tumor dynamics that help define the IS, UT, and LT regions. First, we find that there is enhanced proliferation within the epithelial compartment of the LT region (Fig. 1e, h). Second, we find that there are distinct clinically relevant tumor epithelial states enriched within different parts of the tumor with fewer $LY6D^+$ clones found within the LT region (Fig. 1k)[14]. Third, we find that there are distinct stromal compositions of cells within the different regions (Fig. 1i, j). Specifically, with key importance to this study, we find a larger ratio of HLA-DR$^+$ cells relative to the CD3$^+$ cells within the LT region (Fig. 1j).

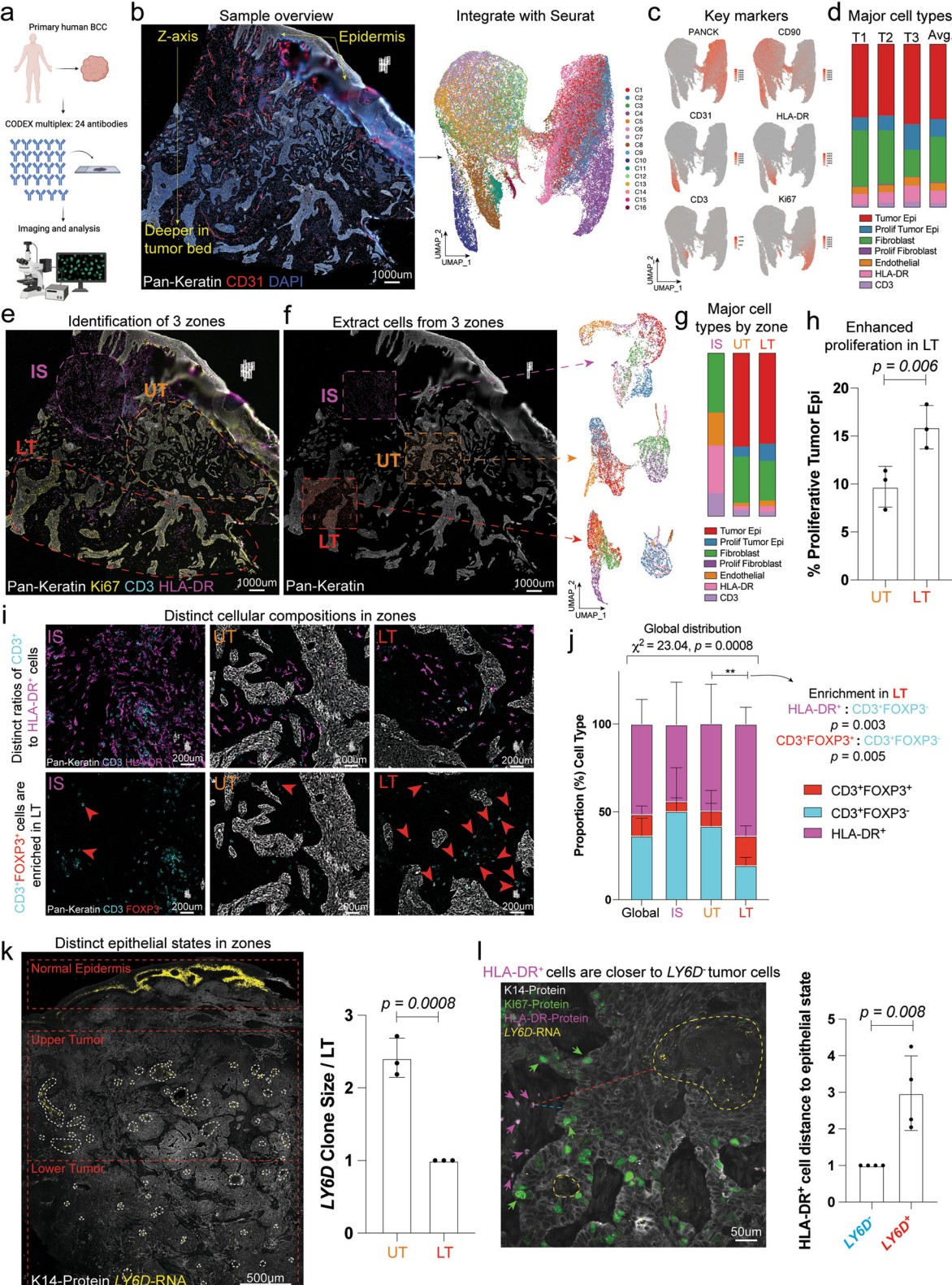

Importantly, these HLA-DR⁺ cells are closer to *LY6D⁻* tumor epithelial cells (Fig. 1l). We subsequently focused our analysis on deconvoluting the identity and role of the HLA-DR⁺ LT-associated cells.

**Identification of an *HLA-DR⁺ITGAM⁺CD68⁺TREM2⁺* macrophage**
To deconvolute the cellular makeup of the HLA-DR⁺ immune cell fractions which more abundant within the proliferation-enriched LT,

we first started by interrogating our patient scRNA-Seq datasets (Fig. 2a). Clustering analysis from 4 different patients revealed that *HLA-DR⁺* cells were composed of roughly three myeloid subtypes, consisting of *CD207⁺* Langerhans cells, *ITGAM⁺CD68^Hi^TREM2⁻* macrophages, or *ITGAM⁺CD68^Hi^TREM2⁺* macrophages (Fig. 2a, b and Supplementary Data 1). These distinct clusters were also observed when we examined the HLA-DR⁺ subset of a single patient (Fig. 2c, d).

**Fig. 1 | BCC stromal spatiality dictates cellular states and cellular compositions.**
**a** CODEX study design for human BCCs. **b** BCC sample overview with the tumor
labeled with Pan-Keratin (white) and CD31 (red). Cell fluoresce metadata for each
marker was extracted and integrated with Seurat. **c** Feature plots of markers used to
identify key populations. **d** Quantification of cell types in **c**. **e** Immune Swarm (IS),
Upper Tumor (UT), and Lower Tumor (LT) in the CODEX sample. The tumor epi-
thelium and epidermis labeled with Pan-Keratin (white), KI67 (yellow), CD3 (cyan),
and HLA-DR (magenta). **f** Cell fluoresce metadata for each marker was extracted
from each distinct neighborhood defined in **e** and integrated with Seurat.
**g** Quantification of cell types by neighborhood region. **h** Quantification of the
proliferative tumor by neighborhood region. **i** Representative images of CD3⁺ cells
(cyan)/HLA-DR⁺ cells (magenta) and FoxP3⁺ (red) cells/CD3⁺ (cyan) cells within
neighborhood regions. Red arrows point to CD3⁺FoxP3⁺ cells. **j** Cellular ratios
between CD3⁺ cells and HLA-DR⁺ cells globally and within the neighborhood
regions. Chi-Square analysis ($X^2$ = 23.04, $p$ = 0.0008, two-tailed) show

neighborhoods do not follow a normal distribution. Fisher Exact analysis (two-
tailed) between the LT and UT for the HLA-DR⁺ to CD3⁺FoxP3⁻ ($p$ = 0.003) and
CD3⁺FoxP3⁺ to CD3⁺FoxP3⁻ ratios indicate nonrandom associations. **k** Spatial
localization and quantification of *LY6D* (yellow) clones within different regions of
K14⁺ tumor epithelium (white). White dotted lines indicate *LY6D*⁺ clones. **l** Spatial
localization of HLA-DR⁺ cells (magenta) relative to the proliferative Ki67⁺ (green)
*LY6D*⁻ tumor epithelium ($n$ = 4 independent tumors). Tumor is labeled with K14
(white), Ki67 (green), HLA-DR (magenta), and *LY6D* (yellow). Green arrows show
KI67⁺ cells. Magenta arrows show HLA-DR⁺ cells. Dotted yellow line indicates *LY6D*⁺
region. Length of each scale bar is noted in figure. For **d**, **g**, **h**, **j**, **k**, $n$ = 3 independent
tumors. For **b**, **e**, **f**, **i**, representative data from a single biological replicate ($n$ = 3
biological replicates collected). Error bars represent mean +/− SD. $p$ values unless
specified otherwise were calculated using an unpaired, two-tailed $t$ test. Source data
are provided as a Source data file.

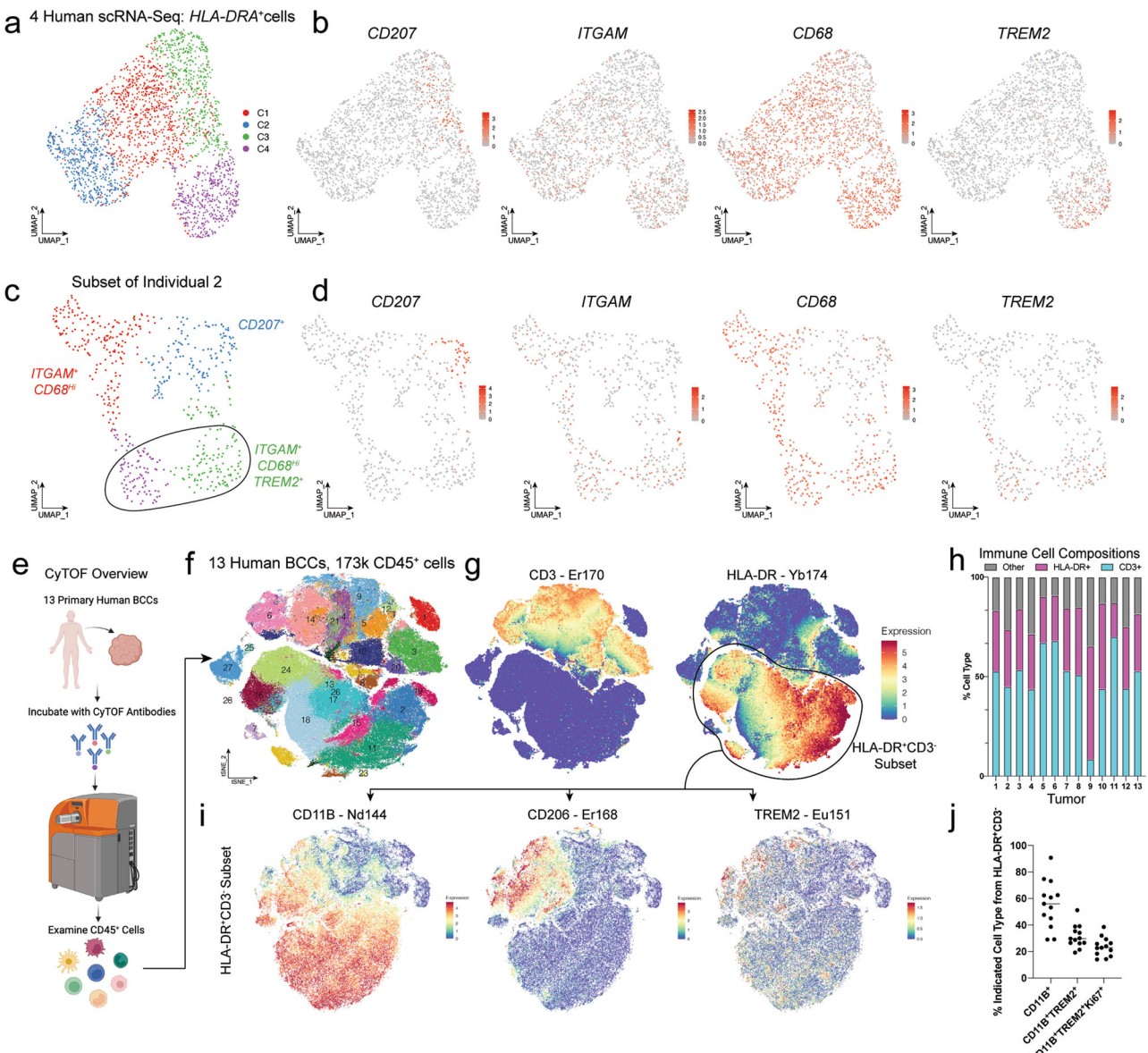

**Fig. 2 | Identification of an *HLA-DR*⁺*ITGAM*⁺*CD68*⁺*TREM2*⁺ macrophage. a** Merged
UMAP plot of scRNA-Seq data of *HLA-DRA*⁺ cells ($n$ = 4 independent tumors).
**b** Feature plots for CD207, *ITGAM*, *CD68*, and *TREM2* from the UMAP that is shown
in **a**. **c** UMAP plot of scRNA-Seq data of *HLA-DRA*⁺ cells from an individual sample.
**d** Feature plots for *CD207*, *ITGAM*, *CD68*, and *TREM2* from the UMAP that is shown
in **c**. **e** CyTOF study design for examining human BCC tumor single-cell suspen-
sions. **f** TSNE plot of the merged CD45⁺ cells from BCC samples. **g** Feature plots for

the key CD3⁺ and HLA-DR⁺ clusters within the CD45⁺ fraction. **h** Quantification of
the proportions of CD3⁺, HLA-DR⁺, and other cell types by sample. **i** Feature plots
for CD11b, CD206, and TREM2 from the subset of HLA-DR⁺ cells. **j** Quantification of
the percentages of cells that are HLA-DR⁺CD11b⁺, HLA-DR⁺CD11b⁺TREM2⁺, and HLA-
DR⁺CD11b⁺TREM2⁺Ki67⁺. For **f**, **h**, **i**, **j**, $n$ = 13 independent tumors. Source data are
provided as a Source data file.

Our discovery that a subset of *HLA-DR*+ cells are *TREM2*+ was of great interest, given the involvement of TREM2+ macrophages in skin homeostasis, neurodegenerative disorders, and immunosuppression in cancer[21,22,29,30]. The expression of *TREM2 is* a predictor of poor clinical outcomes in patients (Supplementary Fig. 2a, b). Furthermore, careful examination of clinical trial data from BCCs, suggests a potential association with enhanced numbers of *TREM2*+ cells in patients that have failed aPD1 immunotherapy (Supplementary Fig. 2c)[18].

We confirmed the presence and features of HLA-DR+TREM2+ cells using cytometry by time of flight (CyTOF) in 13 human BCC tumor samples (Fig. 2e and Supplementary Fig. 2d). Across all samples and like our findings from CODEX, most immune cells were CD3+ or HLA-DR+ cells, with some variation in composition across the tumors (Fig. 2f–h and Supplementary Fig. 2e). Analysis of the HLA-DR+ subset found that most of these cells were myeloid cells of differential polarization status, often co-expressing CD14, CD11B, and CD11C (Fig. 2i and Supplementary Fig. 2e, f). There was variation in the myeloid subsets, which seemed to be driven by a CD33+CD36+CD38+ population and an HLA-DR^Hi^CD206+ population with the highest levels of TREM2 (Fig. 2i and Supplementary Fig. 2f). Surprisingly, TREM2+ cells comprised between 20-50% of the CD11B+ cells present within the tumors (Fig. 2j). Furthermore, we noted that a fraction of the TREM2+ cells were Ki67+, indicating active proliferation within the tumor (a point we will return to) (Fig. 2j). Overall, this analysis deconvolutes the HLA-DR+ cell composition and highlights the existence of a TREM2+ macrophage population in human BCCs.

### The spatial maturation and organization of *TREM2*+ cells span the UT to LT axis

With the identification of multiple HLA-DR+ macrophage types including a TREM2+ population, we wondered whether there was a lineage relation between the different populations. Using a single patient as an example, we subsetted the *CD68*+ macrophages, removing the *CD207*+ Langerhans cells and subjected them to Monocle pseudotemporal analysis (Fig. 3a–d). We found a predicted transition from a *TREM2*- (purple) to a *TREM2*+ (green and blue) state (Fig. 3d). We next asked where these different macrophage populations spatially organized themselves within the BCC tumor. We found that the predicted lineage relationship was also spatially organized, with the *CD68*+*TREM2*- cells enriching in the UT while the *CD68*+*TREM2*+ cells were enriched in the LT (Fig. 3e–g). To understand the spatial localization of *TREM2*+ cells in relation to the different tumor epithelial states, we co-stained for *TREM2* and *LY6D* in human BCCs. Like the results seen when we stained for HLA-DR+ cells in relation to *LY6D*, *TREM2*+ cells were found to be spatially localized near the highly proliferative *LY6D*- cells (Fig. 3h). Overall, we conclude the presence of a spatial axis of myeloid polarization toward a *CD68*+*TREM2*+ state adjacent to the highly proliferative *LY6D*- LT epithelium. This adds a fourth key feature that defines global BCC tumor dynamics.

### Trem2+ cells predominate within the mature myeloid fraction of mouse BCCs

As in human tumors, *Trem2*+ cells were also enriched in the TME around mouse BCCs (mBCC) from our X-ray irradiated *Ptch1*+/-;*K14-creER;p53*^fl/fl^ mouse model, which forms tumors that are histologically similar to sporadic human BCCs[12] (Fig. 4a). We sorted the CD45+ immune cell fraction from these tumors and subjected the cells to scRNA-Seq, finding multiple immune populations (Fig. 4a, Supplementary Fig. 3a, and Supplementary Data 2). Extracting the total myeloid cells, we were able to identify the monocyte populations through markers such as *Ly6c1* before generating a subset of mature myeloid cells (Fig. 4b, c, Supplementary Fig. 3b, c, and Supplementary Data 3). We identified the key marker genes associated with each of the clusters and subsequently plotted key cell-type associated markers

such as *Trem2*, *Itgam*, *Itgax*, *Cd14*, and *Ccr2* (Fig. 4d, e). We descriptively named the myeloid clusters on the basis of these markers (Fig. 4c). As in human BCCs, significant *Trem2* expression correlated with transcriptional and surface expression variability in canonical myeloid lineage markers such as *Itgam* (macrophages), *Itgax* (dendritic cells), *Cd14* (monocytes), and Ccr2 (undifferentiated monocytes) (Fig. 4c, e). We confirmed the overlapping expression of these various myeloid markers by flow cytometry of Trem2+ cells (Fig. 4f).

We next used Monocle analysis to better understand the lineage and maturation progression of the mouse myeloid cells within the TME (Fig. 4g). Based on the markers and position of the purple Trem2^Low^Itgam^Hi^Ccr2^Hi^ and the orange Trem2^Low^Itgax^Hi^Ccr2^Hi^, our prediction is that these are likely more recently recruited myeloid fractions, with the purple Trem2^Low^Itgam^Hi^Ccr2^Hi^ being more macrophage-specific and the orange Trem2^Low^Itgax^Hi^Ccr2^Hi^ being more dendritic-specific (Fig. 4g). Our trajectory would predict that these populations give rise to the red Trem2^Hi^Itgam^Hi^Cd14^Hi^–1 cluster, which can subsequently branch into either the more mature macrophage-specific cell type (green Trem2^Hi^Itgam^Hi^Cd14^Low^) or the more mature dendritic cell-specific cell type (blue Trem2^Hi^Itgax^Hi^–1) (Fig. 4g). Overall, these data support directed Trem2+ intratumoral maturation of myeloid cells within the tumor.

Consistent with the intermediate differentiation state we saw in human tumors, subsequent comparisons of mouse myeloid cells indicate that they existed in a heterogeneous state, exhibiting both M1- and M2-like gene expression programs using a BMDM polarization reference dataset (Fig. 4h)[31]. More extensive pseudotemporal analysis (see "Methods") using an M1 and M2 single-cell open chromatin atlas reference dataset reinforces myeloid cells' highly dynamic, heterogeneous nature within the TME (Supplementary Fig. 3e–j).

To understand the spatial localization of *Trem2*+ cells in relation to the different tumor epithelial states in mice, we co-stained *Trem2* and *Ly6d* in mouse BCCs. Like the results seen in human BCCs, *Trem2*+ cells were found to be spatially localized near the highly proliferative *Ly6d*+ cells (Fig. 4i). Overall, this analysis suggests that a similar population of Trem2+ macrophages exists in both mouse and human tumors.

### Trem2+Vcam1+ skin cancer-associated myeloid cells (SCAMs) are distinct from other non-cancer-associated Trem2+ myeloid cells

As Trem2+ populations are heterogeneous and have been shown to regulate epithelial stem cell functions in normal non-cancer-associated processes, we interrogated the unique properties of Trem2+ macrophages in BCCs[30]. We started more broadly, merging scRNA-Seq datasets of BCC myeloid cells with myeloid cells (dendritic and macrophages) with other datasets that have examined the role of Trem2+ cells from different tissue contexts including normal skin, wounded skin, various cancers, and microglia (Supplementary Fig. 4a, b and Supplementary Data 4)[22,23,30,32,33]. Interestingly, we noted that *Trem2*+ cells from our mouse BCC tumor model clustered most closely with *Trem2*+ cells from other tumor models and not from normal or wounded tissues (Supplementary Fig. 4a–f). To look for skin-specific differences, we merged our BCC myeloid cells with myeloid cells from normal skin tissue regeneration and skin repair after wound healing (Fig. 5a and Supplementary Fig. 4g)[30,32]. We noted that macrophages and dendritic cells from normal tissue homeostasis and during wound healing largely clustered away from cells from Trem2+ BCC macrophages (Fig. 5a). During wound healing, *Trem2* levels were largely restricted to macrophage populations that co-expressed *Itgam* and *Adgre1* (Fig. 5a, b and Supplementary Data 5). In contrast, *Trem2*+ cells from BCCs clustered away from the other datasets and, as previously shown, co-expressed *Itgax*, *Itgam*, and *Adgre1* (Fig. 5a, b).

We identified specific markers and biological processes associated with the different cellular populations, noting the enrichment of *Selenop*, *Rack1*, *Gdf15*, and *Vcam1* in *Trem2*+ macrophages from BCCs

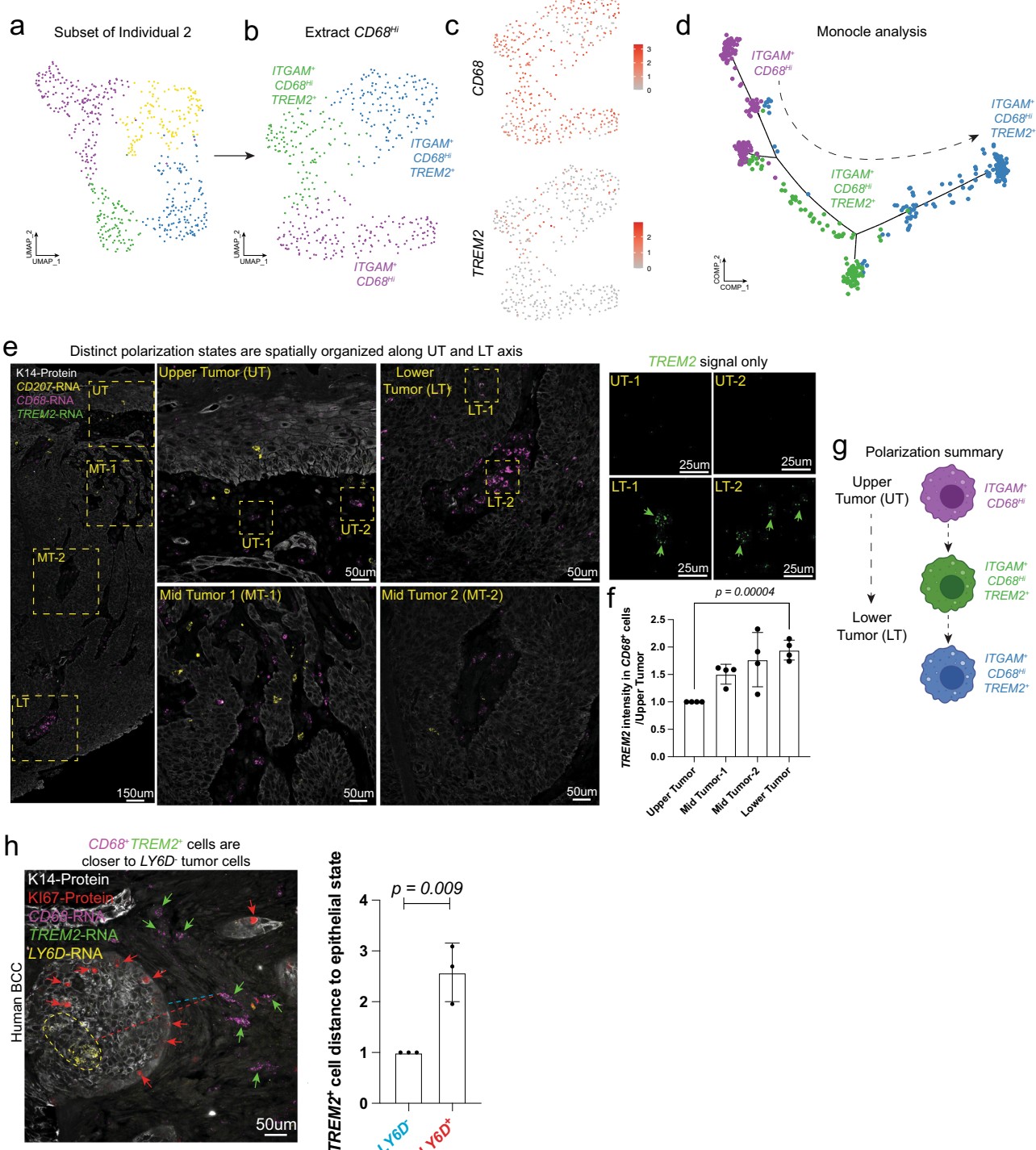

**Fig. 3 | Spatial maturation and organization of *TREM2*+ cells spans the UT and LT axis. a** UMAP plot of scRNA-Seq data of *HLA-DRA*+ cells from an individual sample. **b** UMAP plot of scRNA-Seq data from the *HLA-DRA*+*CD68*^Hi cells from the sample in **a**. **c** Feature plots for *CD68* and *TREM2* from the UMAP that is shown in **b**. **d** Monocle analysis for the 3 clusters from **b**. **e** Spatial locations of *CD68*+*TREM2*+ cells within human BCC tumors via RNAScope for the markers *CD207* (yellow), *CD68* (magenta), *TREM2* (green), and K14-protein (white). Boxes are in different regions of the tumor (UT, MT, and LT). Zoomed in regions of UT and LT are shown with just the *TREM2* (green) signal. **f** Quantification of the *TREM2* (green) signal in the *CD68*+ cells within the different regions of the tumor (*n* = 4 independent

tumors). **g** Macrophage polarization summary diagram. **h** Spatial locations of *CD68*+*TREM2*+ cells within human BCCs relative to the proliferative *LY6D*- tumor epithelium (*n* = 3 independent tumors). RNAScope is for *CD68* (magenta), *TREM2* (green), and protein staining is for K14-protein (white), KI67-protein (red), and *LY6D*-RNA (yellow). Red arrows show some KI67+ cells. Green arrows some show *CD68*+*TREM2*+ cells. Dotted yellow line indicates *LY6D*+ region. Quantification of average cell distances of individual *CD68*+*TREM2*+ cells. Length of each scale bar is noted in figure. Error bars represent mean +/− SD. *p* values were calculated using an unpaired, two-tailed *t* test. Source data are provided as a Source data file.

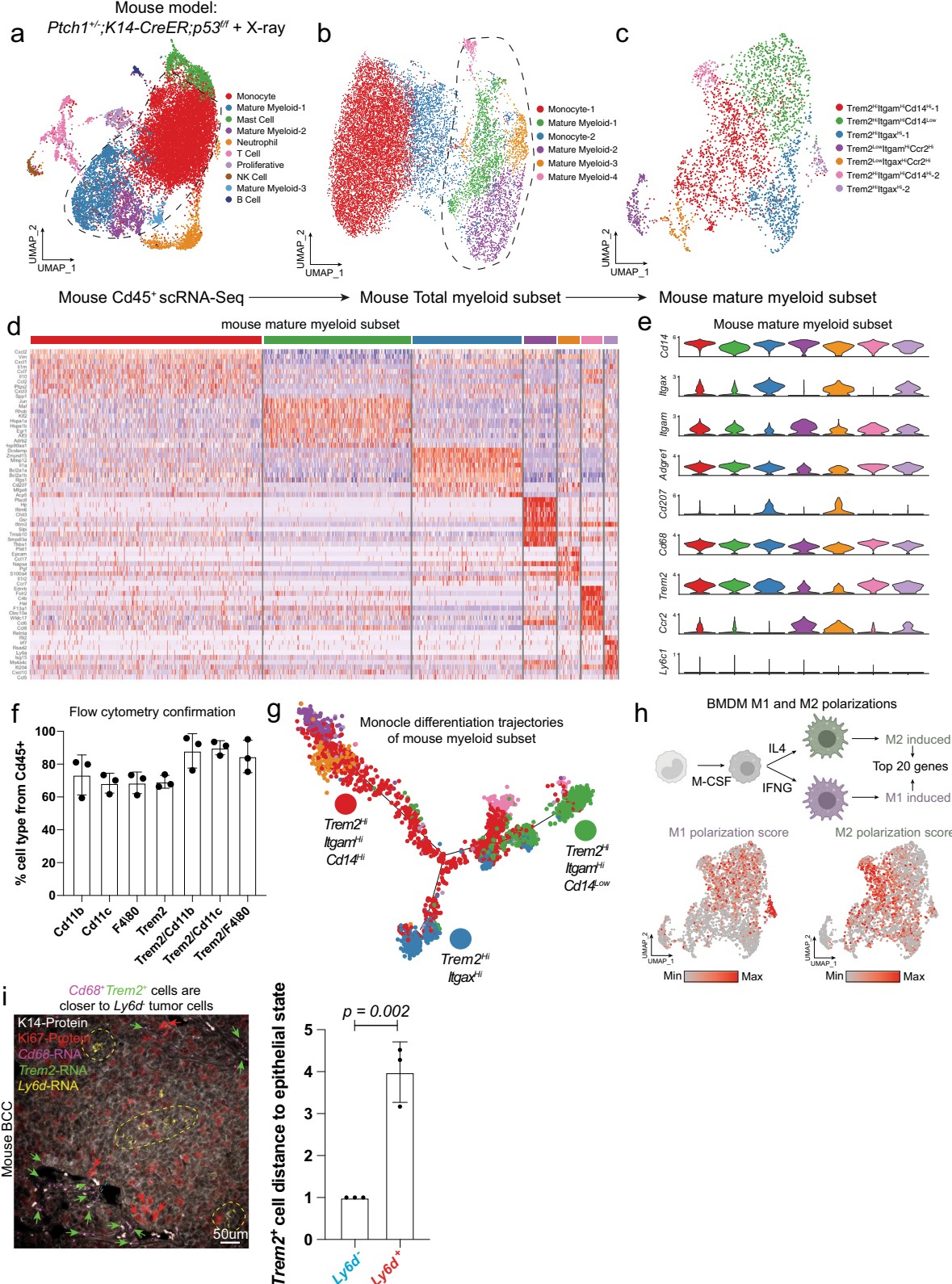

(Fig. 5c, d). We were intrigued by the surface marker Vcam1, which was not induced in an in vitro BMDM system, suggesting specificity to the tumor setting (Fig. 5e and Supplementary Fig. 4i). We confirmed that in both humans and our mouse model, we could detect substantial levels of VCAM1/Vcam1 in the TREM2/Trem2+ fraction (Fig. 5f). Furthermore, we could see co-expression of Vcam*1 and Trem2* by RNAScope (Fig. 5g). We returned to our human CD68+ scRNA-Seq subset and

found that although both the green and blue clusters expressed TREM2, only the terminal blue cluster expressed VCAM1 (Figs. 3b–d and 5h). This provides an additional understanding of spatially-determined lineage transitions seen in human BCCs (Fig. 5i). Subsequent GO analysis of the marker genes of the different populations identified BCC-specific terms related to cell activation, secretion, and immune effector processes, suggesting that these cells are more active

**Fig. 4 | Trem2+ macrophages predominate within the mature myeloid fraction of mouse BCCs. a** UMAP plot of scRNA-Seq data of Cd45+ sorted cells from mouse BCC tumor (*n* = 1 primary tumor). **b** UMAP plot of scRNA-Seq data of myeloid cells from **a. c** UMAP plot of scRNA-Seq data of mature myeloid cells from **b. d** Heatmap showing the top 10 marker genes for the various clusters from **c. e** Stacked violin plot of various key myeloid-associated markers for each of the clusters from **c. f** Quantification of the flow cytometry confirmation analysis of the myeloid cells from mBCC. **g** Monocle analysis of the myeloid cells from **c. h** Diagram of M1 and M2 polarization experiments to generate M1 and M2 polarization scores. Top 20 differential genes between the M1- and M2-polarized cells were extracted and used for scoring. Feature plot of the M1- and M2-associated polarization scoring for clustering from **c. i** Spatial locations of *Cd68+Trem2+* cells within mouse BCCs relative to the proliferative *Ly6d-* tumor epithelium. RNAScope is for *Cd68* (magenta), *Trem2* (green), and protein staining is for K14-protein (white), Ki67-protein (red), and *Ly6d*-RNA (yellow). Red arrows show some Ki67+ cells. Green arrows some show *Cd68+Trem2+* cells. Dotted yellow line indicates *Ly6d-* region. Quantification of average cell distances of individual *Cd68+Trem2+* cells. Scale bar = 50 μm. For **f, i**, *n* = 3 independent tumors. Error bars represent mean +/- SD. *p* values were calculated using an unpaired, two-tailed *t* test. For Source data are provided as a Source data file.

(Fig. 5c). Compared to the other populations, we noted that *Trem2+* cells from BCCs expressed higher proliferative markers such as *Mki67* than macrophages and dendritic cells from wounded and unwounded skin (Fig. 5j). Gene scoring reinforced the GO analysis as we saw higher scoring for both proliferation (a point we will return to) and general macrophage activation (Fig. 5k and Supplementary Fig. 4h). Overall, our analysis indicates that *Trem2+* cells from BCCs have unique gene expression profiles. We refer to these cells in skin cancers as skin cancer-associated myeloid cells (SCAMs).

To better understand the mechanisms regulating SCAM maturation and further characterize key differences between SCAMs and other Trem2+ cells, we used a combination of scATAC-Seq, and bulk ATAC-Seq approaches (Supplementary Fig. 4j–l). Integrating our scRNA-Seq and scATAC-Seq data, we identified undifferentiated monocytes and SCAMs (Supplementary Fig. 4m). At the scATAC-Seq level, we could see peaks in both *Trem2* and *Vcam1* locus largely specific to SCAMs compared to control and wounded skin monocytes (Supplementary Fig. 4n). Lineage prediction tools clearly showed a differentiation transition from monocytes to SCAMs, corresponding with an opening of Trem2-specific peaks suggesting that the transition to a SCAM fate likely occurs within the tumor (Supplementary Fig. 4m, n). Motif analysis along this trajectory found several key motifs, such as the critical myeloid transcription factor PU.1 (Sfpi1), that appeared integral to the transition to the SCAM fate (Supplementary Fig. 4o)[34].

To directly compare the epigenetic state of Trem2+ SCAMs with Trem2+ macrophages from normal skin (tricophages) and wounded skin, we FACS sorted Trem2+ cells from these tissues for ATAC-Seq analysis (Fig. 5l). We found that SCAMs had dramatically different chromatin landscapes than normal and wound macrophages supporting a unique TME epigenetic imprint (Fig. 5m, n and Supplementary Data 6–8). When comparing SCAMs to tricophages, we found that SCAMs were enriched for open chromatin associated with locomotion, cell migration, biological adhesion, and cell proliferation (Fig. 5n, o and Supplementary Data 7). As expected, the *Trem2* locus displayed several open peaks across all samples, but there were some peaks specific to the inflamed wound and tumor context, arguing for the relationship between the tumor and wound environment (Fig. 5p). Importantly, SCAM-specific peaks were also identified, including those within the *Vcam1* locus, further highlighting the utility of Vcam1 for differentiating SCAM populations from other skin Trem2+ cells (Fig. 5p).

Motif analysis indicated that AP-1 family members' binding sites were enriched in SCAMs compared to normal (Fig. 5q), while motifs such as PU.1, SpiB, and AP-1 were lost in SCAMs (Fig. 5q). PU.1 is a critical regulator of monocyte maturation to macrophages, so the loss of PU.1 binding in SCAMs supports a unique differentiation path once they are within the BCC tumor microenvironment (Fig. 5q and Supplementary Fig. 4o). The observation that AP-1 motifs are both enriched and de-enriched in SCAMs compared to normal Trem2+ cells mirrors our previous findings during the BCC to squamous cell carcinoma transition (BST), where subsets of AP-1 binding sites are associated with distinct tumor cell states within the lineage[10]. Furthermore, differential AP-1 binding has been implicated in epigenetic memory of inflammation[35]. Indeed, analysis of AP-1 binding sites between SCAMs

and Trem2+ cells from normal skin reveals the enrichment and de-enrichment of AP-1 motifs at mostly unique sites (Fig. 5r and Supplementary Data 9). Overall, our analysis establishes SCAMs as a tumor-specific Trem2+ population within the *Ly6d-* LT proliferative neighborhood, suggesting a unique role within the TME.

## SCAMs enhance tumor growth

To dissect the contribution of SCAMs within the proliferative *LY6D-* tumor neighborhood, we first wanted to understand the role of the entire tumor stroma. We took advantage of our ability to separate different tumor-associated cell types from our mouse model: tumor epithelium (RFP+) and total tumor stroma (RFP-) (Fig. 6a and Supplementary Fig. 5a). In our allograft model, we found that tumor epithelial cells (RFP+) require stromal cells (RFP-) for continued tumor growth (Fig. 6b, c). Furthermore, using our organoid model, we found that organoid growth was bolstered with the RFP- stroma present (Fig. 6d). This experiment suggests there must be an integral component (or components) in the stroma that drives tumor growth or initiation.

To examine the functional role of SCAMs and test whether they might be essential for tumor growth, we started by depleting myeloid cells using a column-based strategy, either targeting Cd11b+ or Cd11c+ cells, which are both expressed on SCAMs (Fig. 6e). After confirmation of substantial depletion of Trem2+ cells, the remaining cells were allografted into NOD SCID recipients (Fig. 6f and Supplementary Fig. 5b). Cell suspensions depleted of Cd11b+ or Cd11c+ cells generated smaller tumors than controls (Fig. 6g and Supplementary Fig. 5c). We used an antibody neutralization strategy to target Trem2+ SCAMs and test their role in tumor growth after tumor initiation to selectively perturb Trem2+ function after initial tumor formation (Fig. 6h)[22]. We found that the addition of anti-Trem2 antibodies led to a slowing of tumor growth as compared to anti-IgG controls, further supporting the role of Trem2+ cells in tumor growth (Fig. 6i). Addition of anti-Trem2 antibodies led to a reduction in *Trem2* expression of cells as well as reduced numbers of cells when compared to anti-IgG-treated controls (Fig. 6j–l). Furthermore, we noted reduced Ki67+ tumor epithelial cells within the anti-Trem2 treated tumors (Supplementary Fig. 5d). To further validate our findings and provide enhanced human clinical relevance, we utilized a patient-derived organoid (PDO) system where we could culture freshly isolated primary human BCC tumors pieces (Fig. 6m)[36]. The addition of anti-TREM2 antibodies caused a dramatic increase percentage of Apotracker Green epithelial cells in three of the four tumors, suggesting that targeting SCAMs in human BCC samples reduces epithelial cell viability (Fig. 6n, o and Supplementary Fig. 5e). We noted that all three tumors that exhibited an increased level of Apotracker Green within the epithelial cells also showed a reduction in the surface marker expression levels of TREM2 (Fig. 6o, p and Supplementary Fig. 5f). Interestingly, it has recently been reported that addition of anti-TREM2 antibodies could enhance anti-PD1 treatment[22]. We found that in some examples, addition of anti-TREM2 antibodies enhanced the Apotracker Green levels in PDOs that were also treated with anti-PD1 (Fig. 6o).

As SCAMs were primarily localized mainly around the proliferative LY6D- tumor epithelium, we wanted to test tumor epithelial

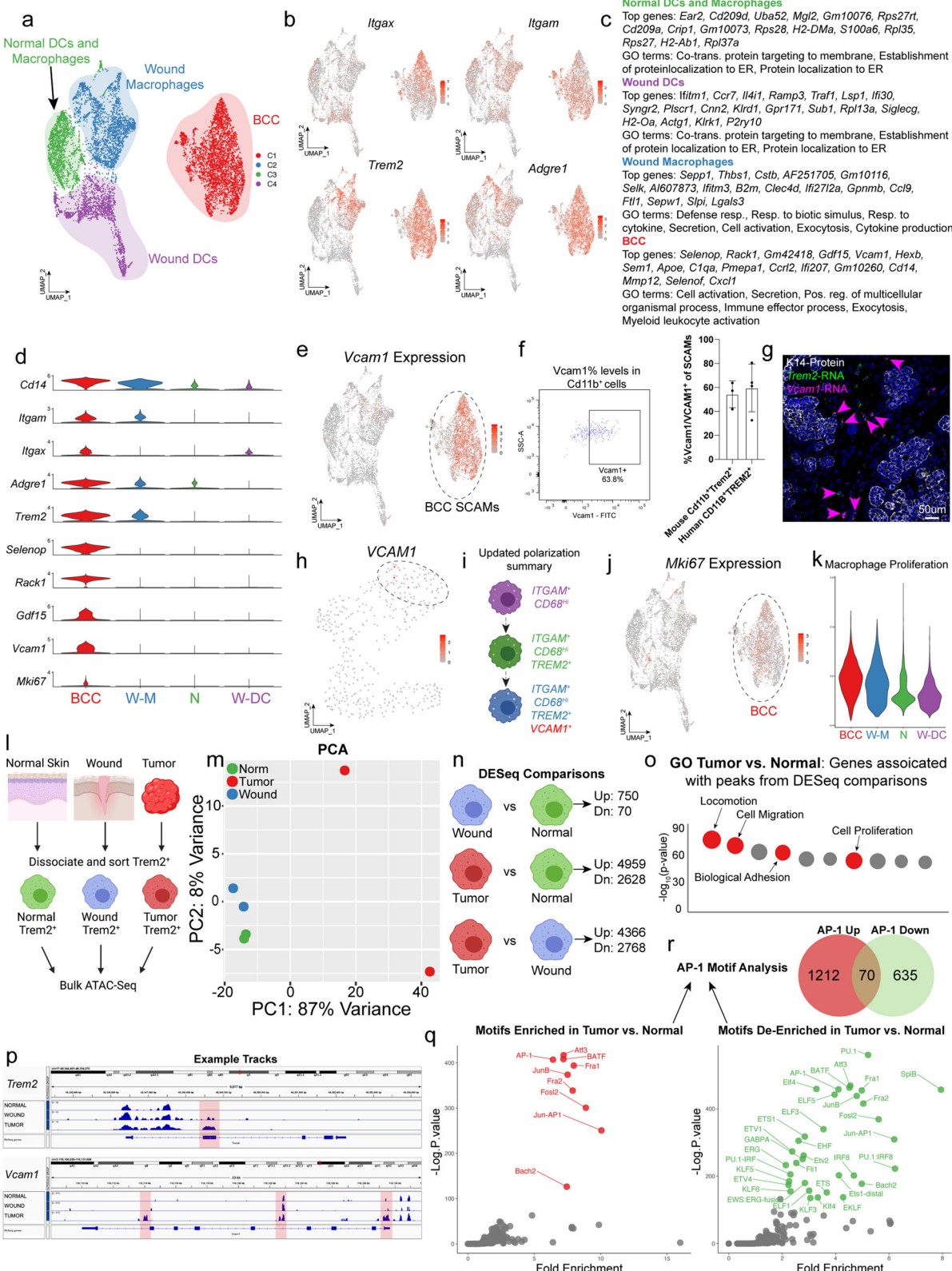

dependence on SCAMs through add-back experiments using our in vitro organoid assay in an LY6D-dependent manner (Fig. 6p and Supplementary Fig. 5f). Like our previous findings, we found that Ly6d+ tumor epithelial cells had better organoid growth potential than Ly6d− cells (Fig. 6q)[14]. We could rescue Ly6d− tumor organoid growth by adding SCAMs, whereas the addition of SCAMs did not affect Ly6d+ organoid growth (Fig. 6q, r). This difference is likely

attributed to the spatial locations of the tumor epithelial populations, with SCAMs being closer to the Ly6d− tumor epithelium (Fig. 4i). Overall, this data provides evidence that SCAMs enhance the growth of specific tumor epithelial populations. Furthermore, these observations highlight a potential direct tumor support mechanism beyond the known roles of TREM2+ macrophages in immunosuppression.

**Fig. 5 | Trem2⁺Vcam1⁺ skin cancer associated myeloid cells (SCAMs) are distinct from other non-cancer associated Trem2⁺ myeloid cells. a** UMAP of merged scRNA-Seq from our myeloid cells from mBCC (*n* = 2 primary tumors), normal skin (*n* = 4 independent samples), and during wound healing (*n* = 3 independent samples). Clusters: normal (green), wounded macrophages (blue), wounded dendritic cells (purple), and BCCs (red). **b** Feature plots for *Itgax*, *Itgam*, *Adgre1*, and *Trem2*. **c** Marker genes and GO analysis associated with **a. d** Stacked violin plots showing marker genes associated with clusters in **a. e** Feature plot of *Vcam1* expression for the scRNA-Seq data in **a. f** Flow cytometry analysis and quantification of VCAM1/Vcam1 in human and mouse BCCs (*n* = 3 independent tumors). Error bars represent mean +/− SD. **g** RNAScope confirmation of *Trem2* (green) and *Vcam1* (magenta) in mBCCs. Magenta arrows indicate dual *Trem2*⁺*Vcam1*⁺ cells. Scale bar = 50 μm. Image is representative of *n* = 2 independent tumors. **h** Feature plot of *VCAM1* expression for the scRNA-Seq data in Fig. 3b. **i** Updated macrophage polarization summary diagram. **j** Feature plot of *Mki67* expression in the scRNA-Seq data in **a. k** Macrophage proliferation scoring of clusters in **a. l** Diagram of the sorting strategies for Trem2⁺ cells from normal skin, wounded skin, and BCCs. **m** PCA plot of the differential bulk ATAC-Seq analysis for Trem2⁺ cells form the different tissues (*n* = 2 samples from each tissue) in **l. n** DESeq comparisons between the Trem2⁺ cells form the different tissues in **l. o** GO terms are associated with the list of genes associated with the differential peaks from DESEq comparisons. *p* values were calculated from the hypergeometric distribution. **p** Example tracks for *Trem2* and *Vcam1* showing open peaks (highlighted in red) that are specific to Trem2⁺ cells from the mBCC. **q** Motifs enriched and de-enriched between SCAMs and Trem2⁺ cells from the normal tissue. Motifs and subsequent *p* values were derived from the HOMER pipeline, which uses cumulative binomial distribution. **r** Venn diagram of the AP-1 motif enrichment/de-enrichment at unique and shared sites that correspond to specific genes. Source data are provided as a Source data file.

## SCAMs promote tumor growth through an OSM-OSMR-STAT3 axis

To determine how SCAMs might promote growth, we utilized our mouse BCC scRNA-Seq data to examine potential receptor/ligand pairs between tumor epithelial cells and SCAMs using CellChat[37]. Initially, we used all cell populations in the scRNA-Seq data and found a highly complex signaling network between all the major cell types within the TME (Supplementary Fig. 6a–f). To focus on the interactions between SCAMs and the tumor epithelium, we subset our data to only include those populations (Fig. 7a, b). CellChat identified several reciprocal signaling modalities between SCAMs and distinct subsets of tumor epithelial cells (Fig. 7c, d and Supplementary Fig. 6g–j). We then focused on SCAM-specific outgoing signals, screening through candidate pathways directed towards the tumor epithelial cells (Fig. 7c). We found that GDF, IGF, OSM, TNF, and TWEAK outgoing signaling pathways were all enriched from SCAMs towards the tumor epithelium (Fig. 7e). Further screening identified the ligands Gdf15, Igf1, Osm, Tnf, and Tnfsf12 as the possible outgoing ligand candidates from the various enriched signaling pathways (Fig. 7f and Supplementary Fig. 6k).

To test the different candidates for tumor growth, we took advantage of our organoid model to screen the other ligands by adding the recombinant ligand to the organoid media (Fig. 7g). We found that the ligand Osm dramatically enhanced organoid growth, generating larger organoids than controls or the other ligands (Fig. 7g, h). Using our scRNA-Seq data, we subsequently confirmed that *Osm* expression was highly specific to SCAMs and that *Osmr* was enriched throughout the tumor epithelium (Fig. 7i). We found greater chromatin accessibility around *Osm* in SCAMs than *Trem2* myeloid cells within the BCC tumors, suggesting that secretion of Osm is associated with a more differentiated state (Fig. 7j). Furthermore, CellChat predicted that the signaling from SCAMs to the tumor epithelium was specifically to the Ly6d- cells, supporting our observations in our SCAM and organoid add-back experiments (Figs. 6r and 7k).

We were surprised by the positive role Osm ligand had on tumor organoid growth given a previous report that Osm-secreting trichophages negatively regulate the highly proliferative hair follicle epithelium[30]. To test the in vivo role of Osm on tumor growth after tumor initiation, we treated tumors with anti-Osm antibodies (Fig. 7l) We found that addition of anti-Osm antibodies dramatically slowed growth of tumors (Fig. 7m). The previous findings that have implicated Osm as a negative regulator of hair follicle stem cell proliferation found that Osm/Osmr signaling acts through JAK-STAT, where trichophages act specifically through phosphorylation of Stat5[30]. We found that within the tumor epithelium, it was Stat3 and not Stat5 signaling that was most abundant, with little to no detection of *Stat5a* (Fig. 7n). We subsequently found that the addition of the Osm ligand was sufficient to induce phosphorylation of Stat3 at the Tyr705 residue suggesting that Osm could induce JAK-STAT3 signaling (Fig. 7o). To test whether downstream JAK-STAT3 signaling could promote tumor growth, we treated our organoids with the small-molecule protein kinase inhibitor ruxolitinib and found dramatically reduced growth (Fig. 7p).

Overall, these results differentiate the mechanism of action of SCAMs from previously identified trichophages, highlighting the context-specific functions mediated through differential Stat activation. Furthermore, these results identify a direct mechanism to promote tumor proliferation, expanding the previously known functions of TREM2⁺ macrophages beyond immunosuppression.

## Tumor assembly of the self-propagating myeloid niche

Compared to myeloid cells found in the skin during tissue regeneration and repair, we noted that SCAMs appear to be actively proliferating within the tumor (Fig. 5j). To this end, we investigated whether SCAMs possessed self-propagating capacity allowing them to maintain themselves within the BCC tumor and enhance tumor growth. To test for the presence of long-term, self-maintaining SCAMs, we first transplanted male primary (Passage 0 or P0) dissociated tumor cells into female NOD SCID mice (Fig. 8a). Two months after tumor formation (Passage 1 or P1), we used RNAScope to look for the male-specific marker *Uty* to differentiate between transplanted and recipient cells. As expected, we saw that the tumor epithelial cells were *Uty*⁺, but interestingly, we could see F4/80⁺*Uty*⁺ indicating the presence of male F4/80⁺ cells within the P1 tumor (Fig. 8b). This observation suggests that even after the pressures of complete dissociation, transplantation, and reformation of a secondary tumor, SCAMs can maintain themselves in the tumor microenvironment.

To further validate and quantify SCAM self-maintenance and rule out the possibility that Nod Scid-specific macrophages are phagocytosing *Uty*⁺ tumor epithelial cells, we took advantage of the ability to discriminate between our tumor mouse (donor), and NOD SCID mouse (host or recipient) strains through the expression of different Cd45 allelic variants (Fig. 8c and Supplementary Fig. 7a, b). P0 tumors, as expected, found that all Cd11b⁺ and Cd11b⁺Trem2⁺ cells were Cd45.2⁺ (Fig. 8d). Interestingly, when we looked at P1 tumors, we found a relatively equal proportion of Cd11b⁺Cd45.2⁺ cells from the P0 mouse and Cd11b⁺Cd45.2⁻ cells contributed by the transplant donor and host (Fig. 8d and Supplementary Fig. 7b). These data (1) confirm previous results above that SCAMs can maintain themselves in the TME and (2) quantifies the proportion of cells that are present (Fig. 8a-d). We also looked at the proportion of cells that were Cd11b⁺Trem2⁺Cd45.2⁺ and found that the proportion was higher relative to the Cd11b⁺Trem2⁺Cd45.2⁻ fraction from the NOD SCID mouse (Fig. 8d). This result may point to potential defects in the NOD SCID myeloid cells (a point we will return to below). We subsequently transplanted P1 tumors into several NOD SCID mice again, generating P2 tumors to test long-term self-maintenance over several months and passages (Fig. 8c). We again found Cd11b⁺Cd45.2⁺ and Cd11b⁺Trem2⁺Cd45.2⁺ cells within the tumors, indicating long-term self-maintenance of SCAMs (Fig. 8d). It is important to note that our transplantation experiments involve allografting one million cells from the P0 or P1

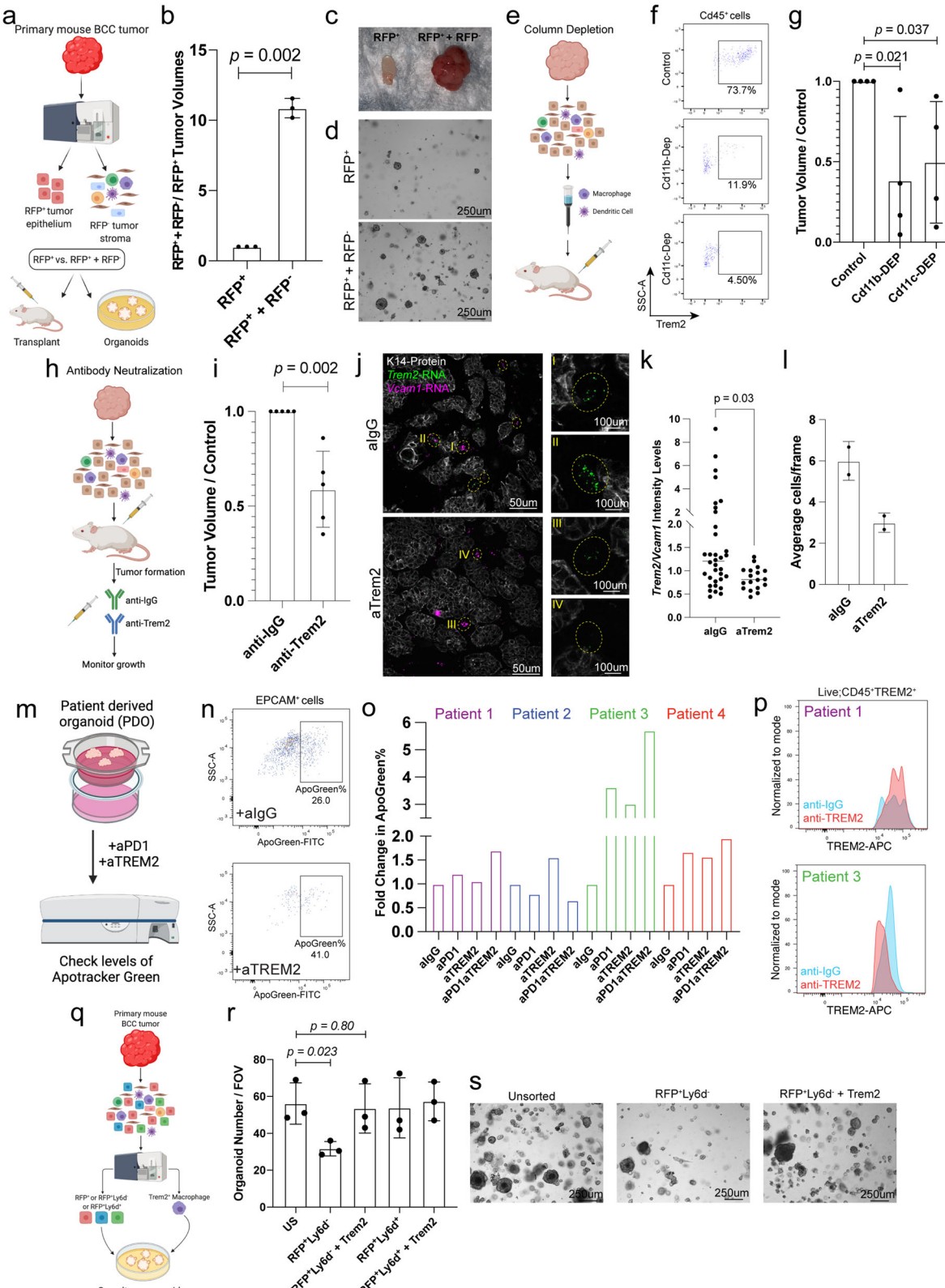

tumor single-cell suspension, which leads to a tumor ranging from 50 to 100 million cells after approximately two months. Even if there are reduced percentages of SCAMs over time due to dilution, the absolute number of Cd45.2+ SCAMs average 1 to 2 logs greater in the P1 or P2 tumors than the absolute number in the one million cells allografted. This point suggests that without any other cellular input of CD45.2+ cells, SCAMs can not only maintain themselves within the tumor, but

they can also expand along with the expansion of the tumor epithelium (Fig. 8e). Overall, these serial transplant assays suggest that the TME generates a highly specialized niche that can drive SCAM self-maintenance.

When we examined the Cd11b+Cd45.2+ fraction of cells in the P0 tumor, we noted a Trem2+ percentage anywhere from 50-75%, but when we looked at the Cd11b+Cd45.2+ in the P1 or P2

**Fig. 6 | SCAMs enhance tumor growth. a** Sorting strategies for RFP+ and RFP− cells and downstream allograft and organoid models. **b** Tumor volumes from allografts (*n* = 3 independent tumors). **c** Images of tumors from transplants in **b**. **d** Representative images of organoid experiments. **e** Diagram of Cd11b+ and Cd11c+ column depletion strategies. **f** Flow cytometry analysis confirming depletion of Cd11b+ and Cd11c+ from tumor fractions before transplanting. **g** Tumor volumes from transplants into NOD SCID mice of control, Cd11b-depleted, and Cd11c-depleted tumor fractions (*n* = 4 independent tumors from 4 mice). **h** Trem2 neutralization strategies to target SCAMs in actively growing tumors. **i** Tumor volumes after Trem2 neutralizations (*n* = 5 independent tumors from 5 mice). **j** RNAScope analysis of *Trem2* (green), *Vcam1* (magenta), and K14-protein (white) in allografted tumors from mice treated with anti-IgG or anti-Trem2 antibodies. Boxes are indicating *Trem2*+ and/or *Vcam1*+ cells with two examples further zoomed in showing only the *Trem2* signal. **k** Quantification of the Trem2/Vcam1 signal in cells in allografted tumors from mice treated with anti-IgG or anti-Trem2 antibodies. **l** Average number of *Trem2*+ and/or *Vcam1*+ cells in allografted tumors from mice treated with anti-IgG or anti-Trem2 antibodies. **m** PDO treatment strategies with anti-TREM2 and aPD1. **n** Flow cytometry analysis showing the levels of Apotracker Green in the EPCAM+ cells. **o** Fold change in ApoGreen percentage between anti-IgG and anti-TREM2 treated PDOs (*n* = 4 independent tumors). **p** Flow cytometry analysis of TREM2 on TREM2+ cells from Patient 1 and Patient 3. **q** Sorting strategies for SCAMs, Ly6d−, and Ly6d+ cells for organoid models. *r* The number of organoids from control and then experimental combinations of the Ly6d− or Ly6d+ tumor epithelial fractions with or without SCAMs (*n* = 3 independent sorts/cultures from 3 independent tumors). **s** Represented images of organoids. Length of each scale bar is noted in figure. For **k, l**, *n* = 2 independent tumors. Error bars represent mean +/− SD. *p* values were calculated using paired, two-tailed *t* test. Source data are provided as a Source data file.

tumors, we noticed a significant increase relative to the quantity of Trem2−Cd45.2+ cells (Fig. 8f, g and Supplementary Fig. 7b). Furthermore, Cd11b+Cd45.2− expressed some Trem2 at lower levels than their Cd11b+Cd45.2+ counterparts (Fig. 8g). This suggests that the ability to self-maintain in the tumor Is Trem2-dependent.

To examine if the Cd45.2+ SCAMs maintain their hallmark features in the P1 tumor, we conducted scRNA-Seq, sorting cells by Cd45.2+ and Cd45.2− cells followed by separate scRNA-Seq library preparations before merging the datasets (Fig. 8h). We noted several unique clusters specific to the Cd45.2+ and the Cd45.2− fractions, likely due to the genetic defects associated with the NOD SCID host myeloid cells (Fig. 8i and Supplementary Data 10). Although Cd45.2− cells could express some *Trem2*, as previously noted through flow experiments, they expressed lower levels (Fig. 8j, k). Furthermore, Cd45.2+ cells maintained higher SCAM-specific features, expressing higher levels of *Osm* and *Vcam1* (Fig. 8j, k).

SCAM self-maintenance appears to be driven by the specialized niche within the TME. Myeloid cells are highly plastic and primarily influenced by their environment, such as in the wound bed during wound repair. During wound healing, monocytes extravasate from blood vessels into the wound bed, subsequently polarizing to distinctly pro-inflammatory or pro-regenerative cell states in a highly orchestrated temporal manner[38]. In the case of SCAMs, we have found that monocyte differentiation into SCAMs occurs within the tumor, suggesting that the tumor can drive SCAM formation (Fig. 4g and Supplementary Fig. 4m). To confirm that monocytes can differentiate into SCAMs within the tumor, we conducted intratumor transplantations of freshly isolated bone marrow-derived monocytes (BMDMs) labeled with GFP (Fig. 8l and Supplementary Fig. 7c)[39]. Five or seven days after transplantation, we found that these injected monocytes expressed substantial levels of both Trem2 as well as the tumor-specific SCAM marker, Vcam1, compared to the monocytes before injection (Fig. 8m, n). The observed levels were comparable to those of the resident GFP− SCAMs suggesting efficient differentiation by the GFP+ BMDMs (Fig. 8n). Overall, this data indicates that the tumor itself can induce SCAM formation.

## Discussion

While ICB therapies have revolutionized cancer treatments, conceptual roadblocks remain to identify additional immune targets that prevent tumor growth[1,2]. Using multiplexed imaging tools coupled with single-cell multi-omic technologies, we characterized the spatially organized immune landscape within the highly proliferative LT region of BCCs. Within the LT region, we observed increased contact between tumor stroma and HLA-DR+ cells and greater numbers of Tregs. Additionally, we found a proliferative TREM2+VCAM1+ macrophage primarily residing in the LT region (which we termed SCAMs) that possesses the surprising capacity to directly regulate LY6D− tumor epithelial proliferation via the secretion of OSM ligand. Previous work on TREM2+ cells has focused mainly on the immunoregulatory properties of myeloid lineages in the tumor microenvironment, emphasizing their role in preventing the activity of tumor-infiltrating effector lymphocytes or regulating the surrounding tumor-associated inflammatory environment[21,22]. In contrast, our work highlights OSM secretion as an immunosuppression-independent mechanism by which SCAMs promote tumor proliferation. We also demonstrate that SCAMs differentiate from myeloid precursors in response to cues within the tumor microenvironment and subsequently self-propagate, maintaining a constant SCAM/tumor epithelium ratio without recruitment and polarization of additional precursors. This work defines a self-propagating proliferative tumor-immune niche, further supporting ongoing efforts to simultaneously target TREM2+ macrophages to enhance ICB-mediated therapies.

A key feature of our work is the spatiality of SCAM function. Our multiplex imaging analysis identifies the local proliferative tumor neighborhoods where the SCAM self-propagating niche operates. Beyond functioning in a spatial-dependent manner within the tumor, we show SCAMs act on a specific tumor epithelial population, promoting LY6D− tumor proliferation that directly contacts the tumor stroma. In contrast, SCAMs appear not to enhance proliferation among LY6D+ tumor cells in more central portions of tumor nodules and segregated away from the tumor stroma. Our previous work suggests that LY6D− tumor epithelial cells are more hair follicle and BCC-like, where LY6D+ tumor epithelial cells are more SCC-like, representing a more differentiated epidermal population[14]. Given that TREM2+ cells exist in SCCs (which have enhanced levels of *LY6D*), we would predict distinct differentiation paths and effector functions of TREM2+ cells between these two tumor types, likely driven by critical differences in the TME and tumor epithelium[40]. Subsequent experiments probing key differences between BCC and SCC TREM2+ cells could reveal shared and distinctive features associated with TREM2+ function and general plasticity associated with myeloid differentiation.

Beyond the functional insights into SCAM function within the TME, we provide an understanding of the unique TME effect on myeloid polarization and differentiation properties. Trem2+ macrophages have been identified in homeostatic, wound repair, and disease conditions, but the differences between non-pathogenic and pathogenic Trem2+ cells have remained unclear[30]. Comparative scRNA-Seq analysis of myeloid cells coupled with ATAC-Seq of sorted Trem2+ macrophages from each tissue aimed to elucidate the similarities and key differences, finding that SCAMs uniquely expressed the surface marker Vcam1 and had an enhanced proliferative capacity. An additional insight was that Osm-expressed within the Vcam1+Trem2+ SCAM niche but not the Vcam1−Trem2+ trichophage niche led to proliferative output. The key difference is derived from differential STAT activation between the tumor (STAT3) and the hair follicle (STAT5) epithelium. These results demonstrate the importance of spatial analysis and the context-dependent signal delivery and reception in understanding microenvironmental function.

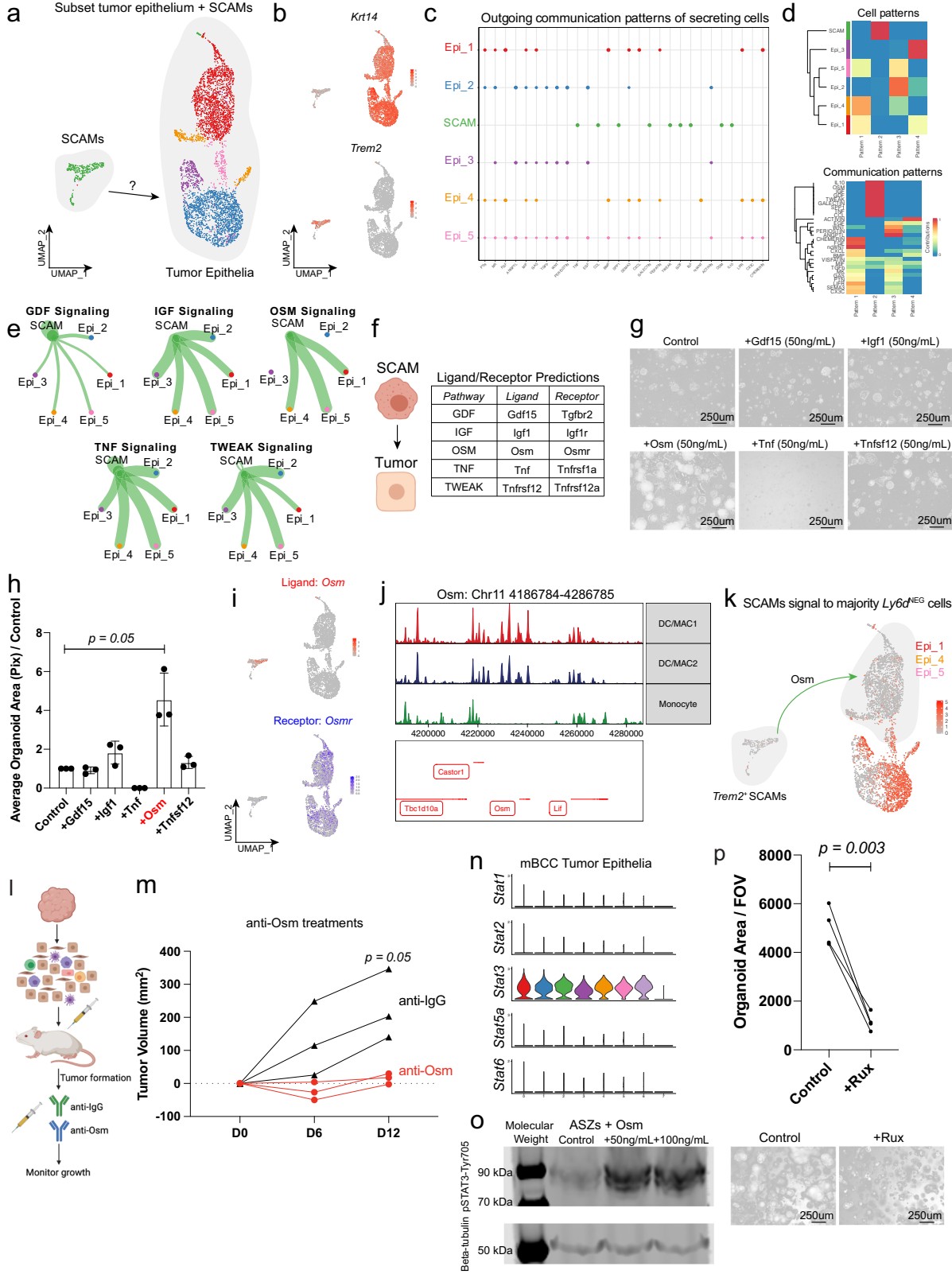

A key feature of our work highlights a key aspect of myeloid biology: the ability to create a self-perpetuating proliferative tumor niche. In line with our observations, recent work has indicated that macrophages contribute to a pro-tumorigenic niche at the early stages of non-small cell lung carcinoma[41]. Drawing parallels to stem cell biology, SCAM induction and maintenance are driven by local

inductive interactions such as between tissue stem cells and organizers, where local inductive interactions facilitate productive stem cell-derived tissue expansion and morphogenesis[42]. These interactions result in a stable and long-term proliferative capacity, and in the context of SCAMs, a transplantable interaction consists of unique and often functionally redundant paracrine signaling. Our work

**Fig. 7 | SCAMs promote tumor growth through an OSM-OSMR-STAT3 axis.**
**a** UMAP of the subset of tumor epithelium and SCAMs for CellChat analysis (*n* = 1 primary tumor). **b** Feature plots for *Krt14* (tumor epithelium) and *Trem2* (SCAMs) in the single-cell data are represented in Fig. 5a. **c** CellChat outgoing communication patters overview. **d** Clustering of the distinct Cell and Communication patterns between the tumor epithelium and SCAMs. **e** Signaling pathways of specific out-going patterns from SCAMs to the tumor epithelium. **f** Candidate list of ligands predicted to be secreted by SCAMs from the signaling pathways in Fig. 5e.
**g** Representative images of organoids treated with predicted ligands from Fig. 5f.
**h** Quantification of the organoid area from Fig. 5g (*n* = 3 independent cultures from 3 independent tumors). **i** Feature plots for the ligand *Osm* and the receptor *Osmr* in the single-cell data are represented in Fig. 5a. **j** scATAC-Seq analysis of the peaks

associated around *Osm*. **k** Predicted signaling interaction between SCAMs and the *Ly6d* fraction of the tumor epithelium. **l** Diagram of Osm neutralization strategies to target Osm ligand in actively growing tumors. **m** Growth curve of tumors treated with anti-IgG or anti-Osm (*n* = 3 independent tumors from 3 independent mice). **n** Stacked violin plots showing expression of the different Stat genes within the mouse BCC tumor epithelium. **o** Western blot for p-STAT3 in mouse ASZ BCC cell line at the Tyr705 residue after addition of Osm (*n* = 2 replicate experiments). **p** Representative images of organoids and subsequent quantification of the orga-noid area after treatment with the Jak/Stat inhibitor Ruxolitnib (*n* = 4 independent tumors). Length of each scale bar is noted in figure. Error bars represent mean +/− SD. *p* values were calculated using unpaired, two-tailed *t* test. Source data are provided as a Source data file.

broadens the immune function in tumor biology, provides a partial explanation for ICB therapy impotence, and nominates the self-propagating tumor immune niche as a potential therapeutic target.

Lineage and spatial localization of SCAMs and serial transplanta-tion studies in NOD SCID mice provide insights into tumor-associated macrophage homeostasis. Previous reverse translational studies have identified enhanced numbers of tumor-associated macrophages after ICB, leading to the assumption that the pool of macrophages depends on the recruitment of precursors from bone-derived monocytes[17,23,25,26]. Our transplantation studies argue that constant influx by functional BMDMs is not required for tumor growth as the transplanted CD45.2[+] SCAMs were able to expand several orders of magnitude within the tumor niche and maintain effector functions without their non-functional CD45.1[+] counterparts. While influx is not required for tumor maintenance, our intratumor injections demon-strate the potent effect on TME-derived SCAM differentiation and polarization within days of microenvironmental instruction. These results identify both a self-replicating and bone-marrow-derived source of SCAMs, providing an important and redundant SCAM source for tumor growth maintenance.

While our experimental results cannot distinguish between TME-derived maturation/ polarization factors for SCAMs, the experi-mental systems developed here will fuel future work to identify the key factors driving this polarization as modulating or preventing SCAM polarization could provide additional therapeutic value. The SCAM polarization and effector function appear spatially localized, suggesting that this tumor-driven polarization may be driving or enriching a particular TME that benefits growth. Along this line, we also noted that Tregs are increased in areas where SCAMs are pre-sent. Although we did not directly test the function of Tregs, it is interesting to speculate not only their role, which has been studied in other cancer contexts but the mechanisms of their recruitment to a particular site and their interactions with SCAMs. Our work does not preclude the involvement of other stromal components within the TME in driving tumor growth. Clearly, additional experiments should involve more extensive numbers of datasets allowing for the better capture of spatial features that might differ between different BCC subtypes. Furthermore, better demarcation methods should also be employed to better define the tumor architecture. Overall, our work highlights the expanding functions of TREM2[+] myeloid cells within the TME and further makes a case for targeting this population to enhance ICB.

## Methods
### Ethics statement
All research presented complies with relevant ethical regulations. All patient samples were obtained through written informed consent and de-identified. All protocols for sample acquisition and usage are reviewed by the Stanford University Institutional Review Board, pro-tocol #18325 (Stanford, CA). All mouse work was approved by the Institutional Animal Care and Use Committee (IACUC) at Stanford University under protocol #11680.

### Human sample processing and isolation
Human tumor specimens were briefly rinsed with 1× PBS before being chopped and minced to pieces less than 1 mm in diameter. After mincing, tumor pieces are transferred to 50-mL conical tube and 40 mL of 0.5% collagenase solution (Gibco; 17-100-017) in DKFSM media (Gibco; 10744-019). The minced tumors are then incubated at 37 °C with rotation for 2 h. At the end of the 2-h incubation, 5 mL of 0.25% Trypsin (Gibco; 25200056) was then added to the 50-mL conical tube, which was then further incubated at 37 °C with rotation for an addi-tional 15 min. 5 mL of FBS was then added to the cellular suspension.

### Mice strains
*Ptch1*[+/−]*;K14-creER;p53*[fl/fl] mice (with or without *RFP*[f-s-f]) are generated as previously indicated[12]. At seven weeks of age, recombination is induced through 3 successive 4-hydroxytamoxifen injections (50μL of 5 mg/mL; Sigma; H7904) completed once per day for 3 days in order to induce recombination of *p53* within the K14-expressing skin epithelia. Within 24 h of the last injection, the mice are X-ray irradiated at 5.25 Gy, leading to loss of the second copy of *Ptch1*, with tumors forming approximately 6 months later. C57BL/6 mice (Stock number: 000664), NOD SCID mice (Stock number: 001303), and C57BL/6-Tg(UBC-GFP) 30Scha/J (Stock number: 004353) were obtained from the Jackson Laboratory.

### Mice experimental methods
For mouse wounding experiments, we wounded as previously indicated[32]. Briefly, female mice at 7 weeks of age were anesthetized using isoflurane (VETone; 13985-528-60), back shaved, and then a 6-mm punch (Integra; 33-36) was used to generate two full-thickness wounds. Wounds were collected 4 days later. For tumor allografting experiments, we passaged as previously indicated[12]. Allografting occurs when we take primary (P0) tumor single-cell suspensions and allograft into Nod Scid mice, which will form a secondary (P1) tumor. Serial allografting of secondary (P1) into Nod Scid mice can also occur to form tertiary (P2) tumors. Briefly, 1 million cells were suspended 50 μL of a 50:50 mix of DFKSM and Matrigel (Corning; 356237) from dissociated mouse tumors were injected subcutaneously into NOD SCID mice. For mouse tumor depletion experiments, single-cell sus-pensions were incubated with either CD11b (Miltenyi Biotec; 130-126-725) or CD11c (Miltenyi Biotec; 130-125-835) MicroBeads UltraPure at appropriate concentrations as per manufacturers recommendations. After incubation, cell suspensions were passed through LD Columns (Miltenyi Biotec; 130-042-901) as per manufacturers' recommenda-tions. After passing through the columns, cell suspensions were pel-leted, counted, and then prepared for 1 million cell injections by methods mentioned above into NOD SCID mice. Control, CD11b-depleted, and CD11c-depleted suspensions were all injected into dif-ferent places on the same NOD SCID mice. For anti-Trem2 injection experiments, 1 million cells from the same bulk tumor were injected by methods mentioned above into paired NOD SCID mice. Once the tumors were palpable, initial measurements were made before pro-ceeding with antibody injections. For paired NOD SCID mice, we

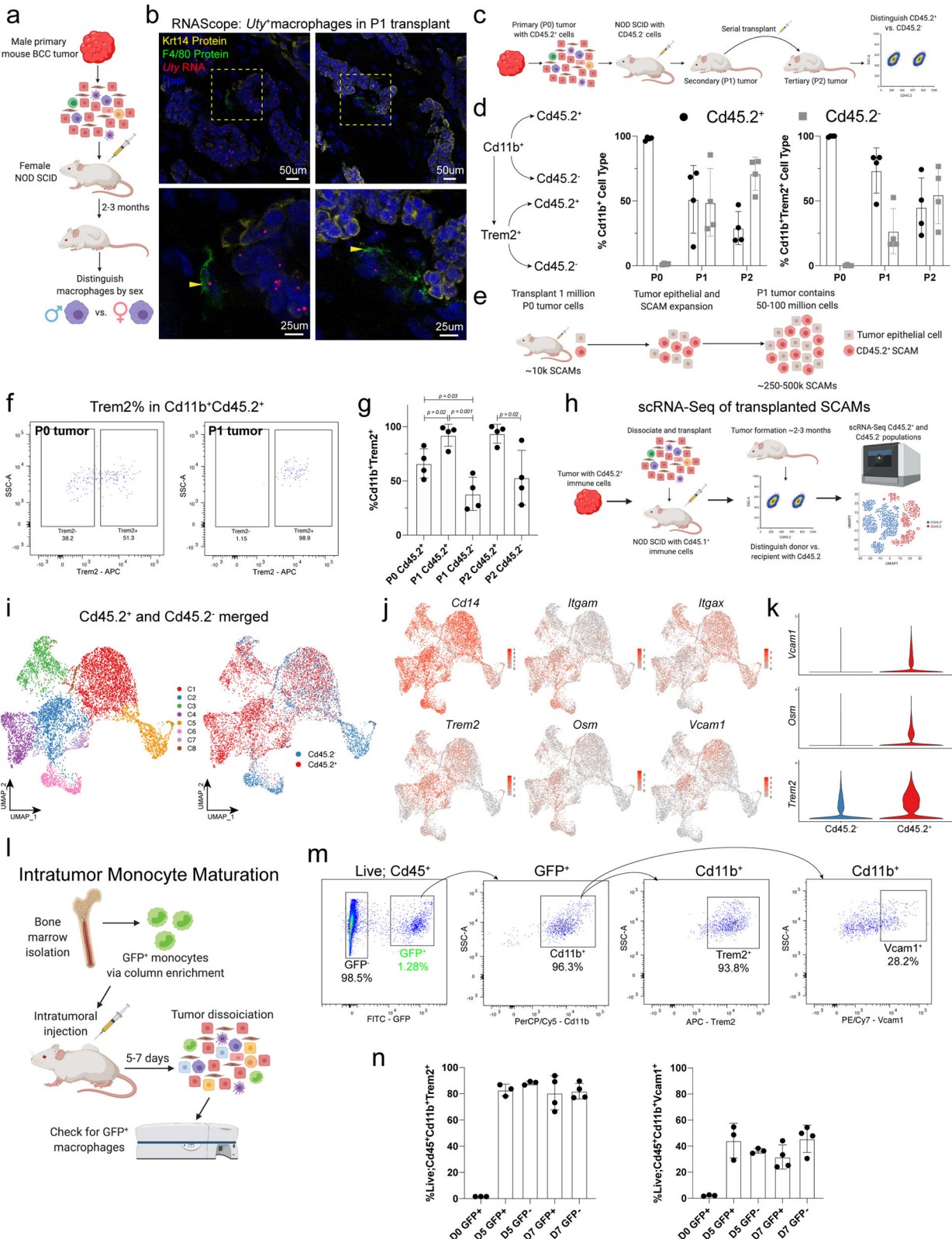

injected 200 µg of anti-Trem2 or anti-IgG antibody (gifts from the Marco Colonna lab) every day for 5 days[22]. Measurements were made two weeks after the first injection. For Osm neutralization experiments, 1 million cells from the same bulk tumor were injected by methods mentioned above into paired NOD SCID mice. Once the tumors were palpable, measurements were made, and mice were then injected with 20 µg of anti-Osm (R&D; AF-495-NA) or anti-IgG

(BioXCell; BP0086) antibody for three initial days, followed by every other day for seven total injections. For intratumoral monocyte injections, we followed the protocol as previously indicated[39]. As opposed to growing tumor cells in vitro, we took advantage of our ability to allograft primary tumor cells and prepared for $4 \times 1$ million cell injections by methods mentioned above into NOD SCID mice. Once tumors were at least 500mm³, we prepared GFP⁺ monocytes for

**Fig. 8 | Tumor assembly of the self-propagating myeloid niche. a** Diagram of transplantation strategies of male primary tumor into female NOD SCID mice. **b** RNAScope images of transplants showing male-specific *Uty*[+] (red) F4/80[+] (green) macrophages within the tumor. Yellow arrows point to F4/80[+]*Uty*[+] cells. Image is representative of *n* = 2 independent tumors. **c** Diagram of transplantation strategies and the ability to trace SCAMs from primary tumor using the CD45.2 allelic variant. **d** Summary of the proportions for the CD45.2[−] and Cd45.2[+] from either Cd11b[+] (left) or Cd11b[+]Trem2[+] cells in the P0, P1, and P2 tumors. **e** Diagram summarizing self-propagating logic with approximate cell numbers. **f** Flow cytometry for Trem2 cells in the Cd11b[+]Cd45.2[+] fractions from P0 and P1 tumors. **g** Quantification of the Trem2% of the Cd11b[+]Cd45.2[+] and Cd11b[+]Cd45.2[−] form the P0, P1, and P2 tumors.

**h** Diagram of the transplantation strategy and subsequent sorting to isolate Cd45.2+ and CD45.2- cells from the P1 tumors for scRNA-Seq. **i** UMAP plots showing the different clusters from the Cd45.2+ and CD45.2- cells. **j** Feature plots showing key genes associated with SCAMs. **k** Violin plots showing key genes associated with SCAMs. **l** Diagram of the intratumor monocyte maturation injection strategy and subsequent analysis by flow cytometry. **m** Flow cytometry analysis of the injected GFP[+] monocytes for SCAM-specific markers. **n** Quantification of the SCAM-specific markers Trem2 and Vcam1 (*n* = 4 independent injections into 4 different tumors). Length of each scale bar is noted in figure. For **d**, **g**, *n* = 4 tumors from P0, P1, and P2 allografts. Error bars represent mean +/− SD. *p* values were calculated using unpaired, two-tailed *t* test. Source data are provided as a Source data file.

injection from the bone marrow of C57BL/6-Tg(UBC-GFP)30Scha/J mice as described, including usage of the Monocyte Isolation Kit (Miltenyi Biotec; 130-100-629) as per the manufactures instructions[39]. $5 \times 10^5$ GFP[+] monocytes isolated from different C57BL/6-Tg(UBC-GFP)30Scha/J mice were injected into the 4 different tumors on the NOD SCID mice. At 5 or 7 days after injection of the GFP[+] monocytes, we processed our tumors using our in-house methods for tumor dissociation as described below. Mice were fed regular chow, fed water ad libitum, and maintained under a 12-h light/dark cycle at 20−24 °C with 42% humidity in accordance with our mouse protocol. Mouse monitoring followed all experimental-dependent protocols. In accordance with our mouse protocol, the maximal tumor size/burden is 1.75 cm³ in volume. In our experiments, the maximal tumor size/burden was not exceeded. All mouse euthanasia followed a carbon dioxide-based euthanasia followed by cervical dislocation as described in our protocol.

## Mouse sample processing and isolation

Mouse tumor (mBCC) specimens were processed similarly to human tumor specimens. Excised mouse tumors were chopped and minced to pieces less than 1 mm in diameter before being transferred to 15-mL conical tubes containing 10 mL aliquots of 0.5% collagenase solution (Gibco; 17-100-017) in DKFSM media (Gibco; 10744-019). The tumors were incubated at 37 °C with 600 rpm shaking for 1 h. At the end of the 1-h incubation, 1.5 mL 0.25% Trypsin (Gibco; 25200056) was added to each 10 mL aliquot, which was then incubated at 37 °C with 600 rpm shaking for an additional 10 min. The mixture was then neutralized with 10% FBS in DMEM (Gibco; 11995-065) before being filtered with a 70-μm filter and then pelleted. Cells were either immediately used for various experiments or frozen in BAM Banker (FujiFilm; CS-02-001). For isolation of normal mouse epidermis, 7-week-old female C57BL/6J mice were shaved, back skin removed, and underlying fat was scrapped off. The skin was then minced into pieces smaller than 1 mm in diameter. Minced samples were placed in 15-mL conical tubes and digested with 10 mL of 0.25% collagenase solution in DKFSM. Samples were incubated at 37 °C with rotation for 2 h. 1 mL of FBS was then added to the cellular suspension. The dissociated cellular suspensions were filtered with a 70-μm filter before being pelleted and immediately used for downstream experiments. For isolation of wound samples from mice, back skin was removed 4 days after wounding, and then a 10-mm punch biopsy (Acuderm; P1050) was used to punch out the wound region. Once the wound sample was isolated, it was processed similarly to the normal epidermis methods mentioned above.

## CODEX imaging

For CODEX antibody conjugation and validation, conjugation utilized all protocols and reagents per manufacture instructions (Akoya; 7000009). For conjugations, 50 μg of purified, carrier-free FOXP3 antibody (ThermoFisher; 14-5773-82) was used and conjugated to BX042-Cy5 (Akoya; 5350008). Centrifugation was performed at $12,000 \times g$ for 8 min for all steps unless otherwise stated. Antibodies were concentrated on a 50kDa filter equilibrated with filtration buffer.

The sulfhydryl groups were activated by incubating for 30 min at room temperature (RT) with the reduction mix. The antibody was then washed with conjugation buffer once. The oligonucleotide barcode was resuspended in the conjugation buffer, added to the antibody, and allowed to incubate for 2 h at RT. The conjugated antibody was washed 3 times, by resuspending and spinning down with purification solution. Antibody was then eluted by adding 100 μL storage buffer and spinning at $3000 \times g$ for 4 min. The conjugated antibodies were stored at 4 °C till use. Before use in the multiplex CODEX experiments, conjugated antibodies were validated with CODEX single stains on human fresh frozen BCC tissue. Staining was performed with the conjugated antibody as described below. The screening buffer was prepared according to the CODEX manual provided by Akoya Biosciences. The fixed and stained tissue was incubated in the screening buffer for up to 15 min. Fluorescent DNA probe was prepared and added to the stained tissue for 5 min. The tissue was washed 3× with screening buffer followed by 1× CODEX buffer. The tissue was then imaged using a Keyence BZ-X810 inverted fluorescent microscope. All CODEX antibody staining of fresh frozen tissue was done according to Akoya Biosciences staining protocol associated with the CODEX Staining Kit with some modifications (Akoya; 7000008). After tissue hydration, autofluorescence was bleached using a previously published protocol by allowing the tissue to sit in a solution (made of sodium hydroxide and hydrogen peroxide) sandwiched between a LED light panel for 45 min at RT[43]. The tissue was then washed twice in ddH₂O for 10 min, followed by a wash in CODEX Hydration Buffer (contained in CODEX Staining Kit) twice, 2 min each. The coverslip was allowed to equilibrate in CODEX staining Buffer (contained in CODEX Staining Kit) for 20 min at RT. The blocking buffer was prepared by adding N, S, J, and G blocking solutions to CODEX staining buffer (contained in CODEX Staining Kit). All antibodies were used at a 1:50 dilution. Antibodies were added to this blocking buffer to make a total volume of 100 μl. The antibodies used were CD2-BX002-550 (Akoya; 4250005), CD3-BX015-Cy5 (Akoya; 4350008), CD4-BX021-Cy5 (Akoya; 4350010), CD5-BX024-Cy5 (Akoya; 4350011), CD7-BX025-488 (Akoya; 4450022), CD8-BX004-488 (Akoya; 4150004), CD9-BX028-488 (Akoya; 4450016), Cd11c-BX027-Cy5 (Akoya; 4350012), CD19-BX003-Cy5 (Akoya; 4350003), CD21-BX013-488 (Akoya; 4450009), CD31-BX032-550 (Akoya; 4250009), CD34-BX035-550 (Akoya; 4250020), CD38-BX007-488 (Akoya; 4150007), CD45-BX001-488 (Akoya; 4150003), CD69-BX041-550 (Akoya; 4250022), CD89-BX030-Cy5 (Akoya; 4350017), CD90-BX022-488 (Akoya; 4150021), CD104-BX005-550 (Akoya; 4250008), CD138-BX010-488 (Akoya; 4150008), CD278-BX017-550 (Akoya; 4250013), CD279-BX014-550 (Akoya; 4250010), HLA-DR-BX026-550 (Akoya; 4250006), KI67-BX047-550 (Akoya; 4250019), PANCK-BX019-488 (Akoya; 4150020). The antibody cocktail was added to the coverslip, and staining was performed in a sealed humidity chamber at 4 °C overnight. After staining, coverslips were washed twice in hydration buffer for 4 mins followed by fixation in storage buffer (contained in CODEX Staining Kit) with 1.6% paraformaldehyde for 10 min. Coverslips were then washed thrice in PBS, followed by a 5 min incubation in ice-cold methanol on ice for 5 mins followed by another 3 1× PBS washes. CODEX fixative solution

(contained in CODEX Staining Kit) was prepared right before the final fixation step. 20 µL of CODEX fixative reagent was added to 1000 µL 1× PBS. 200 µL of the fixative solution was added to the coverslip for 20 mins followed by 3 washes in 1× PBS. Coverslips were then immediately prepped for imaging. For CODEX multicycle setup and imaging, coverslips were mounted onto Akoya's custom-made stage between coverslip gaskets with the tissue side facing up. The coverslips were cleaned from the bottom using a Kim wipe to get rid of any salts. The tissue was stained with CODEX Nuclear Stain (Akoya; 7000003) at a 1:2000 dilution in 1× CODEX buffer. A 96-well plate was used to set up the multicycle experiment with different fluorescent oligonucleotides in each well. A reporter stock solution was prepared to contain 1:150 Hoechst stain and 1:12 dilution of assay reagent in 1× CODEX buffer (contained in CODEX Staining Kit). Fluorescent oligonucleotides (Akoya Biosciences) were added to this reporter solution at a final concentration of 1:50 in a total of 250 µL per well. A blank cycle containing no fluorescent probes was performed at the start and end of the experiment to capture autofluorescence. Automated image acquisition and fluidics were performed using Akoya's software driver CODEX Instrument Manager (CIM, version v1.29), and the CODEX platform (Akoya Biosciences). Imaging was performed using a Keyence BZ-X810 microscope, fitted with a Nikon CFI Plan Apo λ ×20/0.75 objective. The BZ-X software (for Keyence microscope) multi-point option was used to define the center and the imaging area corresponding to each region. 9 z steps were acquired with the pitch set at 1.5 in the BZ-X software. Raw tiff files were processed using the CODEX Processor version 1.7.0.6 by Akoya Biosciences. The software allows you to perform drift-compensation, deconvolution, and background subtraction using the blank cycles and stitching of the large images. Segmentation was performed using the nuclear stain signal to generate comma-separated values (csv) and flow cytometry standard (fcs) files containing the spatial coordinates of each detected cell and the signal intensities for all the markers for each region.

**CODEX analysis**
CODEX files were imported into CODEX Multiplex Analysis Viewer (MAV) for analysis as described (https://help.codex.bio/codex/mav/overview). Isolation of cells (global samples, cells from IS, cells from UT, and cells from LT) for subsequent analysis in Seurat was done as described using the CODEX STEP Analysis (https://cdxstpnlns-mv21-17d1fc.netlify.app/). After clustering was complete from the CODEX STEP Analysis, markers from the CODEX panel were then used to identify key cell types. To determine the major cell types (seen in Fig. 1d) as well as the finer resolution analysis of the neighborhoods (shown in Fig. 1f, g), we used the Seurat "table" function to identify the number of cells in each cluster from the global samples, which correspond to a particular cell type. For the differential analysis between the HLA-DR+, CD3+FOXP3-, and CD3+FOXP3+ (shown in Fig. 1k), we had to utilize fluoresce thresholds to identify cell types as CD3+FOXP3- and CD3+FOXP3+ cells because although distinct visually, they clustered mainly with one another. We used the Seurat "subset" function, where we could subset the cells based on a threshold of fluoresce intensity. For the HLA-DR+ cells, we used a threshold of >5000. For the CD3+FOXP3- and CD3+FOXP3+ cells, we first used a threshold of >2500 for CD3 and then >2500 for FOXP3.

**scRNA-Seq library preparation and sequencing**
As per the directions of the Chromium Single Cell 3′ Reagents Kit v2 (following the CG00052 Rev B. user guide), mouse live FACS-sorted cells were washed in PBS containing 0.04% BSA and resuspended at a concentration of approximately 1000 cells/µL. We aimed to capture 10,000 cells. Each library was sequenced on the Illumina NovaSeq platform (using the HiSeq and NextSeq System Suite software) to achieve an average of roughly 30,000 reads per cell.

**scRNA-Seq library processing, quality control, and clustering**
FASTQ files were aligned using 10× Genomic Cell Ranger 3.1.0 and aligned to an indexed mm10 for mouse samples. Standard workflow using Seurat v3 (within R v4.2.1) with quality control parameters of 200-5000 gene detection and mitochondrial percentage under 10% were utilized to filter cells. Cells were presented in two-dimensional space using UMAP with the following sample specific parameters and subset workflow listed below. For analysis of the human HLA-DR+ samples, CD45+HLA-DRA+ cells were extracted from four human BCC samples (two previous datasets with two new datasets) and clustered using the top 3 PCs with a resolution of 0.3[14]. A single sample was then subsetted for subsequent analysis and clustered using the top 3 PCs with a resolution of 0.3. Cells were initially clustered for the mouse Cd45+ sample using the top 15 PCs with a resolution of 0.4. Clusters that were not *Ptprc*+ (such as contaminating *Krt14*+ epithelial cells or *Col1a2*+ fibroblasts) were removed, and the sample was re-clustered using the same above parameters. SCAMs were isolated by their expression of *Trem2*, *Itgam*, and *Itgax* from the Cd45+ sample and merged with SCAMs from the previous mBCC sample[14]. The merged sample was clustered using the top 10 PCs with a resolution of 0.3. To analyze myeloid cells from normal skin, wound, and SCAMs, we merged our SCAM dataset with myeloid datasets from previous publications and were clustered with the top 8 PCs[30,32]. For the Cd45.2+ and CD45.2- experiments, all myeloid cells were isolated by their expression of *Cd14*, *Itgam*, and *Itgax*. These subsets were then merged and clustered using the top 10 PCs with a resolution of 0.3.

**scRNA-Seq analysis**
Marker genes were found for each cluster using the standard Seurat pipeline and parameters such as log(fold-change) >0.25 as a cutoff for performing differential gene expression. For gene scoring, we used the AddModuleScore function built into Seurat. Gene Ontology analysis was conducted using marker genes for the various clusters, which were submitted to MSigDB collection from the Broad Institute. For gene scoring of M1 and M2 polarization states, we used the top20 genes that define M1(IFNG) and M2(IL-4) polarization states from a mouse bone marrow-derived macrophages reference scRNA-seq dataset[31]. To construct a pseudotemporal view on polarization programs, we utilized a scATAC-seq reference dataset that established pseudotime trajectories of M1(IFNG) and M2(IL-4) polarization in mouse bone marrow-derived macrophages. Using the top20 genes that exhibited significant chromatin accessibility changes (based on their accessibility−GeneScore) in a pseudotemporal manner (accessibility lost−lost; accessibility gained early by the 33rd percentile of the trajectory−early, accessibility gained late by the 66th percentile of the trajectory−late) in the single-cell chromatin states of M1 and M2 macrophages, we scored and visualized the enrichment of this signature in the SCAM BCC scRNA-seq dataset. For Monocle-based lineage analysis, clusters and variable genes information is pulled from the human patient subset or mouse SCAM Seurat object and taken as input to construct pseudo time trajectory using the semi-supervised Monocle 2 (2.14.0) workflow. For CellChat (1.1.3) analysis, we used the CellChat tutorial and vignette (https://github.com/sqjin/CellChat) as the basis of all analysis based on the original publication[37]. For the global mBCC sample, we used five outgoing patterns. For the mBCC sample containing only the tumor epithelium and SCAMs, we used four outgoing patterns.

**scATAC-Seq library preparation and sequencing**
As per the directions of the Chromium Next GEM Single Cell ATAC Library & Gel Bead Kit v1.1, single-cell suspensions of mBCCs were FACS-sorted for live CD45+ cells were subsequently washed in PBS containing 0.04% BSA and resuspended at a concentration of approximately 1000 cell/µL. We aimed to capture 10,000 cells. Each library was sequenced on the Illumina NovaSeq platform to achieve an average of roughly 30,000 reads per cell.

## scATAC-Seq library processing, quality control, and clustering

For the scATAC-Seq library processing, the "cellranger-atac count" function (cellranger-atac, v1.1.0) is used for pre-processing of the 10x-based scATAC-seq data using mm10 mouse genome. scATAC-Seq analysis was carried out using the R package ArchR (v1.0.1). Within ArchR, "Arrow files" are generated for each sample from the fragment file with the parameters of 1000 unique fragments and a transcription-start-site enrichment value of 4. An "ArchRproject" was created for a merged dataset where all arrow files from 2 mouse samples. We used the "addIterativeLSI" function of ArchR to perform dimensionality reduction on each ArchRproject and applied Harmony algorithm to mitigate batch effects for the merged ArchRproject. For clustering, the "addClusters" function was used followed by plotting using UMAP with the "addUMAP" and the "plotEmbedding" function. To identify cells of interest, we used the Trem2 GeneScoreMatrix. We created pseudo-bulk replicates for each cluster and called peaks using MACS2, identified peaks unique to each cluster, and used "getMarkers" function to obtain the marker list of each cluster. We than added motif annotation to ArchRproject, identifying enriched motifs associated with differential peaks. We applied the "addImputeWeights" function to impute the weights of marker features and the "plotEmbedding" function to visualize the marker features. We performed pairwise comparisons of peaks and TFs between clusters using the getMarkerFeatures function and plotted an MA or volcano plot using the "markerPlot" function in ArchR. "addCoAccessibility" function is used to examine co-accessibility which returned a loop track that represented the co-accessibility information using the "getCoAccessibility" function. Finally, "plotBrowserTrack" function is used to plot genome browser tracks of peaks and co-accessibility. Later, we defined the trajectory backbone of cell groups or clusters based on myeloid/SCAM state. We then created a trajectory using the "addTrajectory" function and plotted the pseudotime values on UMAP embedding using the "plotTrajectory" function.

## ATAC-Seq

Bulk ATAC-Seq experiments were conducted as previously indicated[44]. Briefly, 50,000 FACS sorted Trem2+ cells were sorted and then pelleted at 500 RCF at 4 °C for 5 min. Cell pellet was resuspended in 50 µL of ATAC-Resuspension Buffer (RSB) containing 0.1% NP40 (catalog), 0.1% Tween-20 (catalog), and 0.01% Digitonin (catalog) then incubated on ice for 3 min on ice. Wash with 1 mL of cold RSB containing 0.1% Tween-20 but not NP40 or Digitonin. Nuclei were then pelleted at 500 RCF at 4 °C for 10 min. The supernatant was removed, and the cell pellet was resuspended in 50 µL of transposition mixture as previously described[44]. The reaction was incubated at 37 °C for 30 min in thermomixer 1000 RPM. The reaction was cleaned up with the QIAquick PCR Purification kit (Qiagen; 28104) and then amplified for 8-11 cycles to generate libraries for sequencing as per protocol[44]. Briefly, libraries were sequenced initially on Illumina MiSeq to determine enrichment score via calculation of the signal over background at the transcriptional start site (TSS) over a 2-kb window. Libraries with the highest score were subsequently chosen for deep sequencing on the Illumina NovaSeq platform. Paired-end reads were trimmed, processed, and mapped to mm10 using Bowtie2 2.3.4.1 with parameters -p 4−very-sensitive[13,45]. Samtools v1.8 was used to discard duplicate reads, narrow peaks were called using MACS2 with the parameters−nomodel−extsize 73−shift −37 and FDR threshold 0.01, and background removal was carried out via submitting replicates to irreproducible discovery rate (IDR) filtering[46,47]. The DESeq2 v1.38.3 package was used to determine differentially accessible peaks from a union peak list generated from overlapping peaks[48]. Peaks and motifs were annotated using the Homer suite v4.11[49]. Gene Ontology analysis was conducted using genes from the differential peaks, which were submitted to MSigDB collection from the Broad Institute.

## RNAScope

All RNAScope experiments were performed using the Multiplex Fluorescent v2 system (ACD; 323100) in conjuction with the RNA-Protein CO-Detection Ancillary Kit (ACD; 323180) as per manufacture protocols. Briefly, human and mouse tumors were fixed with 4% par-aformaldehyde overnight at 4 °C, paraffin-embedded, and sectioned at 5 µm. We followed the formalin-fixed, paraffin-embedded (FFPE) protocol as per manufacturing instructions. The following human probes were used: Hs-CD207 (ACD; 809521-C3), Hs-CD68 (ACD; 560591-C2), Hs-TREM2 (ACD; 420491-C1), Hs-TREM2 (ACD; 420491-C3), Hs-LY6D (ACD; 484681-C1), Mm-Trem2 (ACD; 404111-C3), Mm-Vcam1 (ACD; 438641-C2), and Mm-Ly6d (ACD; 532071), Mm-Cd68 (ACD; 316611-C2). For protein staining, the following antibodies were used: chicken K14 antibody (1:500; BioLegend SIG-3476-100), rabbit Ki67 antibody (1:100; Abcam; ab16667), and mouse HLA-DR (1:100; Abcam; ab136320). The various secondary antibodies were used: anti-chicken Alexa488 secondary antibody (1:500; Invitrogen; A-11039), anti-chicken Alexa405 secondary antibody (1:500; Invitrogen; A48260), anti-rabbit Alexa488 (1:500; Invitrogen; A21206), and anti-mouse Alexa555 (1:500; Invitrogen; A31570). Nuclei visualization, when done, was with Hoechst 33342 (1:1000; ThermoFisher; 62249). Imaging was conducted using a Lecia SP8 with the Lecia LAS X v3.7.6 used for data collection. Quantification of RNAScope signal was completed using FIJI 2.

## Flow cytometry/fluorescence activated cell sorting (FACS)

For FACS or flow experiments, single-cell suspensions of cells were resuspended in a 2% FBS solution and then stained with appropriate antibodies at a 1:100 dilution for 60 min at 4 °C in the dark. Antibodies used were APC anti-human/mouse TREM2 (R&D; FAB17291A), APC/Cy7 anti-mouse CD45 (BioLegend; 103116), Alexa700 anti-mouse CD45.2 (BioLegend; 109822), PerCP/Cy5 anti-mouse CD11b (BioLegend; 101228), BV605 anti-mouse Cd11c (BioLegend; 117334), FITC anti-mouse Ly6d (BioLegend; 138606), PE/Cy7 anti-mouse Vcam1 (BioLegend; 105720), BC510 anti-mouse F4/80 (BioLegend; 123135), PE/Cy7 anti-human CD326 (BioLegend; 324222), and PE anti-human VCAM1 (BioLegend; 305805). Before sorting or flow, SytoxBlue (Thermo Fisher; S34857) was used for a viability dye at a 1:1000 dilution. FACS experiments were run on FACSAria II, and flow experiments were run on LSRII instruments, where BD FACSDiva v8.0.1 was used for data collection. Subsequent analysis was done in FlowJo v10.6.

## Organoid model

Organoid model was based off mouse epidermal and hair follicle organoid culture and our previously published mouse BCC organoid model[14,50]. Briefly, single-cell suspensions of bulk tumor (from our Ptch1+/;p53f/f;K14Cre-ER;RFPf-s-f mouse model), or specific FACS sorted mouse tumor epithelial cells were plated at 10,000 cells per 10 µL BME (Trevigen; 2432-005-01) drop in a 6-well plate with eight drops per well or a 24-well plate with two drops per well. Cells were cultured in EEM as previously described. Briefly, the contents include: Advanced DMEM/F12 (Gibco; 12491015) supplemented with penicillin/streptomycin (100 U/L; Gibco; 15140122), Hepes, (10 mM; Gibco; 15630080), GlutaMAX (1×; Gibco; 35050061), 1× B27 supplement (50x stock from Gibco; 17504044), N-Acetylcysteine-1 (1 mM; Sigma Aldrich; A9165-5G), Noggin-conditioned medium (5%, derived from HEK293-mNoggin-Fc cells, provided by Hans Clevers), R-spondin 1-conditioned medium (5%, derived from HEK293-HA-R-Spondin1-Fc cells; R&D Systems; 3710-001-01), acidic FGF1 (100 ng/mL; PeproTech; 100-17A), Heparin (0.2%; Stemcell Technologies; 7980), Forskolin (10 ng/mL; Tocris; 1099), Rho kinase inhibitor (Y-27632; 10 µM; Stemcell Technologies; 72302), and 1× Primocin (500x stock; InvivoGen; ant-pm-1). We also found that adding TGF-β inhibitor SD208 (1µM; Tocris; 3269) enhanced growth. HEK293-mNoggin-Fc cells (provided by Hans Clevers) and HEK293-HA-R-Spondin1-Fc cells (R&D Systems; 3710-001-01) used to generate

conditioned media cells were monitored by their morphology. No mycoplasma was detected through regular testing. The media was replenished every three days. For SCAM add-back experiments, 1000 SCAMs were added to the 10,000 tumor epithelial cells before plating. For the experiments testing potential ligands for growth effect on tumor epithelial organoids, acidic FGF1 was removed from the media. The recombinant ligands Gdf15 (R&D; 8944-GD), Igf1 (R&D; 791-MG), Osm (R&D; 495-MO), Tnf (R&D; 410-MT), Tnfsf12 (R&D; 1237-TW-025) were added at a concentration of 50 ng/mL to the culture media. Organoids grew for 14 days and were subsequently imaged for counting and size measurements. For drug treatment, organoids were treated with Ruxolitinib (1 μM; MedChemExpress; HY-50856).

## Patient-derived organoid (PDO) model

The PDO model was based previous established protocols for culturing pieces of tumor and culturing them long term[36]. Briefly, Millicell-CM inserts (MilliporeSigma; PICM03050) were prepared with 1 mL of an 8:1:1 ratio of Cellmatrix Type 1-A (Fujifilm; 637-00653), 10× concentrated culture media HAM's F-12 (catalog), and reconstitution buffer. Small pieces of human tumor were re-suspended in 1 mL of Cultrex Reduced Growth Factor Basement Membrane Extract, Type R1 (Biotechne; 3433-010-R1) and then placed on the solidified 8:1:1 collagen matrix. 1 mL of PDO media in Advanced DMEM/F-12 (Gibco; 12634010), supplemented with: 50% WNT3A, RSPO1, and Noggin-conditioned media (ATCC; CRL-3276), Hepes, (1 mM; Gibco; 15630080), GlutaMAX (1×; Gibco; 35050061), N-Acetylcysteine-1 (1 mM; Sigma Aldrich; A9165-5G), penicillin/streptomycin (100 U/L; Gibco; 15140122), nicotinamide (10mM; MilliporeSigma; N3376-100G), B-27 Supplement (50×) minus vitamin A (1x; ThermoFisher; 12587010), A83-01 (0.5 μM; Tocris; 2939), Gastrin I human (10nM; MilliporeSigma; G9020-250UG), SB202190 (10 μM; MilliporeSigma; S7067-5MG), Recombinant human IL-2 (1000 units/mL; Peprotech; 200-02), Human EGF (50 ng/mL; ThermoFisher, PHG0311L), Recombinant Human M-CSF (20 ng/mL; Peprotech; 300-25). L-WRN cells (ATCC; CRL-3276) used to generate conditioned media cells were monitored by their morphology. No mycoplasma was detected through regular testing. PDOs were treated with with Human IgG4 antibody (10 μg/mL; R&D Systems; MAB9895-100), TREM2 antibody (10 μg/mL; R&D Systems; MAB17291-500), and/or PD1 antibody (10 μg/mL; Selleckchem; A2002) for 7 days before collecting samples for analysis. After the 7 days, the tissues from each well were further dissociated into single cells using 5 mL of 0.25% collagenase solution solution (Gibco; 17-100-017) in DKFSM media (Gibco; 10744-019) incubated at 37 °C with rotation for 30 min to first dissociate the matrix that the PDOs are suspended in. PDOs are pelleted, the 0.25% collagenase solution removed and then replaced with 5 mL of a 0.5% collagenase solution incubated at 37 °C with rotation for 1 h. At the end of the 1-h incubation, 0.5 mL of 0.25% Trypsin (Gibco; 25200056) was then added to further dissociate for an addition 10 min. Apoptosis of the epithelial fraction was assessed with Apotracker Green (Bioloegend; 427402).

## CyTOF sample preparation and acquisition

All CyTOF experiments were performed using the Maxpar Nuclear Antigen Staining with Fresh Fix (400277 Rev06) protocol, allowing surface marker and nuclear staining. Vials of single-cell dissociated cells containing 1–3 million cells were thawed and spun down at 300 × g for 5 min in 10% FBS solution in PBS to remove residual freezing media. Cells were then washed again in PBS and spun down at 300 × g for 5 min. Cells were then re-suspended with 200 μL of PBS. Before surface antibody staining, we performed viability staining with Cell-ID Cisplatin-198Pt (Fluidigm; 201198). A 2× working stock of Cell-ID Cisplatin-198Pt was made by adding 2 μL of Cell-ID Cisplatin-198Pt with 998 μL of PBS. 200 μL of the 2× working stock was then added to each sample that had been re-suspend with 200 μL of PBS. Samples were incubated for 5 min at room temperature. Wash twice 5 mL of

Maxpar Cell Staining Buffer (Fluidigm; 201068), spinning samples down at 300 × g for 5 min. Re-suspend cells in 50 μL of Maxpar Cell Staining Buffer. Prepare the surface marker antibody cocktail using the Antibody Cocktail Preparation Guide contained within the Maxpar Nuclear Antigen Staining with Fresh Fix protocol. The antibodies used from the Maxpar Human Mono/Macro Phenotyping Panel Kit (Fluidigm; 201317) were: CD19-142Nd, Cd11b-144Nd, CD7-147Sm, CD66-149Sm, CD36-152Sm, CD163-154Sm, CD11c-159Tb, CD14-160Gd, CD16-165Ho, CD38-167Er, CD206-168Er, CD33-169Tm, CD3-170Er, HLA-DR-174Yb. Additional surface marker antibodies were CD45-89Y (Fluidigm; 3089003B), CD207-175Lu (Fluidigm; 3175016B), CD279-155Gd (Fluidigm; 3155009B), CD8a-146Nd (Fluidigm; 3146001B), TIM-3-153Eu (Fluidigm; 3153008B), TIGIT-209Bi (Fluidigm; 3209013B), and CD274-156Gd (Fluidigm; 3156026B). TREM2 antibody was conjugated in house using the Maxpar X8 Antibody Labeling kit for 151Eu (Fluidigm; 201151A). All antibodies were used at the 1× concentration (1:50) as suggested by protocol expect CD33-169Tm, CD279-155Gd, and TREM2-151Eu, which were used at 2× (1:25), 2× (1:25), and 4× (1:12.5), respectively. Add 50 μL of the antibody cocktail to the 50 μL suspension of cells and mix. Incubate the sample at room temperature for 30 min but gently vortex sample at 15 min to re-suspend again. Wash twice 5 mL of Maxpar Cell Staining Buffer, spinning samples down at 300 × g for 5 min. With completion of the surface marker staining, the cells are then preprepared for nuclear antibody staining. 1 mL of Maxpar Nuclear Antigen Staining Buffer (Fluidigm; 201063) was added to each sample, briefly vortexed, and then incubated at room temperature for 30 min. After the 30 min, two washes with 2 mL of Maxpar Nuclear Antigen Staining Perm (Fluidigm; 201063) were completed with samples spun down at 800 × g for 5 min. Prepare the nuclear marker antibody cocktail using the Antibody Cocktail Preparation Guide contained within the Maxpar Nuclear Antigen Staining with Fresh Fix protocol. The nuclear antibodies used are Ki-67-172Yb (Fluidigm; 3172024B) and FoxP3-162Dy (Fluidigm; 3162011A). Add 50 μL of the antibody cocktail to the 50 μL suspension of cells and mix. Incubate the sample at room temperature for 30 min. After the 30 min, two washes with 2 mL of Maxpar Nuclear Antigen Staining Perm (Fluidigm; 201063) were completed with samples spun down at 800 × g for 5 min. The cells are then fresh fixed with 1 mL of 1.6% formaldehyde solution (Thermo Scientific; 28906), incubated for 10 min at room temperature, and then spun down at 800 × g for 5 min. The cells are then re-suspended in 1 mL of Cell-ID Intercalator-Ir (Fluidigm; 201192B) at 1:4000 in Maxpar Fix and Perm (Fluidigm; 201067) and left overnight. The next day, cells are washed with 5 mL of Maxpar Cell Staining Buffer followed by two washes of Maxpar Water (Fluidigm; 201069), spinning cells down at 800 × g for 5 min. Samples are run on CyTOF 2 with samples diluted with 0.1× EQ Four Element Calibration Beads (Fluidigm; 201078).

## CyTOF sample analysis

After samples had been run, they were normalized to beads before fcs files were exported. The fcs files were then imported into FlowJo v10.6 for cell selection, live/dead discrimination, and isolation of CD45+ immune cells. CD45+ populations were then subsequently gated, isolating a CD3- but HLA-DR+ population, before being exported as fcs files. These CD45+CD3-HLA-DR+ cells were analyzed using Cytofkit[51].

## ASZ cell culture

ASZ001 BCC cells were cultured in 154CF medium (ThermoFisher; M154CF500) supplemented with 2% chelated FBS and 0.05 mM CaCl₂ as previously indicated[13]. Experiments were carried out using low serum conditions with 154CF medium containing 0.2% chelated FBS and 0.05 mM CaCl₂. ASZ_001 cells were derived from mouse basal cell carcinoma cells and previously authenticated[52]. In our experiments, cells were monitored by their morphology, sequencing, and utilization

of mouse-specific primers. No mycoplasma was detected through regular testing.

## Western blot

On ice, RIPA buffer was added to ASZs on a 10-cm plate, scrapped, and then incubated for 30 min. After lysis, samples were prepared through the addition SDS sample buffer (SDS loading buffer: 6× Laemmli SDS Sample buffer; Bioland Scientific; SAB03-01) and denatured by heating at 95 °C for 15 min. Protein and protein ladder (Chameleon Due Pre-stained Protein Ladder; LI-COR; 928-60000) was then loaded into a precasted SDS gel (4–12% Bis-tris Precasted Gel NUPAGE; NP0321BOX) and ran in MOPs solution (NUPAGEl NP0001). Protein was transferred and then blocked with 5% milk at room temperature for 1 h. Blots were incubated with either P-Stat3 Tyr705 (Cell Signaling; 9145T; 1:1000) or Tubulin (DSHB; 4A1; 1:1000) overnight. Bands were visualized after staining with IRDye anti-Rabbit IgG (LI-COR; 926-68071; 1:10,000) and IRDye anti-Mouse IgG (LI-COR; 926-32210; 10000) on the Odyssey DLx.

### TREM2 expression analysis in human cancer data

To examine the expression levels of *TREM2* in various tumor types as well as look at survival data, we utilized the GEPIA web interface (http://gepia.cancer-pku.cn/).

## Reporting summary

Further information on research design is available in the Nature Portfolio Reporting Summary linked to this article.

## Data availability

The human specific sequencing generated from this study has been deposited in the dbGaP database under the accession code phs003242.v1.p1. All new sequencing data have been deposited in GSE204952. Previous human BCC scRNA-Seq data used in this study are available in dbGAP database under the accession code phs003103.v1.p1. Previous mouse BCC scRNA-Seq data used in this study are available in the NCBI Gene Expression Omnibus under the SuperSeries code: GSE186184. The resistant human BCC scRNA-Seq publicly available data used in this study are available in the NCBI Gene Expression Omnibus under the accession code: GSE123814. Previous mouse colon cancer scRNA-Seq data used in this study are available in the NCBI Gene Expression Omnibus under the code: GSE165404. The mouse microglia scRNA-Seq data used in this study are available in the NCBI Gene Expression Omnibus under the SuperSeries code: GSE140511. The mouse normal skin scRNA-Seq data used in this study are available in the NCBI Gene Expression Omnibus under the accession codes: GSE142471 and GSE108709. The mouse wound scRNA-Seq data used in this study are available in the NCBI Gene Expression Omnibus under the accession code: GSE142471. The mouse sarcoma scRNA-Seq used in this study are available in the NCBI Gene Expression Omnibus under the accession code: GSE151710. The data relating to TREM2 expression in various cancer types can be accessed through the GEPIA web interface (http://gepia.cancer-pku.cn/). Source data are provided with this paper. The remaining data are available within the Article, Supplementary Information or Source data file. Source data are provided with this paper.

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

## Acknowledgements

Cell sorting/flow cytometry analysis for this project was done on instruments in the Stanford Shared FACS Facility. The following NIH Shared Instrument Grants were used to fund equipment used in the Stanford Shared FACS Facility: NIH S10RR025518-01 and NIH S10RR027431-01. Imaging was conducted at the Stanford University Cell Sciences Imaging Core Facility. The following NIH Shared Instrument Grant was used to fund equipment used in the Stanford University Cell Sciences Imaging Core Facility: NIH 1S10OD010580. The work is funded by the following grants: D.H. is supported by NIH 5T32AR7422-37 and NIH 1F32CA254434; T.F. is supported by Stanford MSTP and the Knight-Hennessy Scholarship; J.B. is supported by NSF GRFP; A.R.J. is supported by NIH T32AR007422-40; N.Y.L. is supported by the Singapore A*STAR graduate fellowship National Science Scholarship; A.T.S. was supported by the National Institutes of Health (NIH) K08CA23188-01, U01CA260852, a Career Award for Medical Scientists from the Burroughs Wellcome Fund, a Technology Impact Award from the Cancer Research Institute, an ASH Scholar Award from the American Society of Hematology, the Pew Charitable Trusts Foundation, the Cancer Research Institute's Lloyd J. Old STAR program, and the Parker Institute for Cancer Immunotherapy; A.E.O. is supported by NIH 1R01AR04786 and NIH 2R37ARO54780. The diagrams shown in Figs. 1a, 2e, 3g, 4h, 5i, l, n, 6a, e, h, m, q, 7f, l, and 8a, c, e, h, l and Supplementary Fig. 3e were created with BioRender.com. We thank the Marco Colonna lab at Washington University in St. Louis for their gracious gift of anti-IgG and anti-Trem2 antibodies.

## Author contributions

D.H., A.T.S., and A.E.O. conceived, designed, and interpreted experiments. D.H. and A.E.O. wrote the manuscript. D.H., C.P., T.F., A.R.J., F.G., N.Y.L., and T.P. executed all wet lab-based experiments. D.H., B.D., S.G., and J.B. conducted all computational experiments. Y.C. and J.H. generated the anti-Trem2 and anti-IgG antibodies. S.A. provided all human clinical samples. All authors revised the manuscript.

## Competing interests

A.T.S. is a founder of Immunai and Cartography Biosciences and receives research funding from Allogene Therapeutics and Merck Research Laboratories. The remaining authors declare no other competing interests.
