## [Peer Review File · Nature Communications]

Skin basal cell carcinomas assemble a pro-tumorigenic spatially organized and self-propagating Trem2+ myeloid nicheREVIEWER COMMENTS

Reviewer #1 (Remarks to the Author): with expertise in tumor associated macrophages

In this study, authors explore the spatial organization of the tumor microenvironment of human basal cell carcinoma by using multiplexed imaging and unbiased high dimensional analysis. Authors uncover a specific subset of tumor associated macrophages which express TREM2 and VCAM1 and which co-localize with a highly proliferative population of tumor epithelial cells (LY6D-). By a series of elegant in vivo experiments and organoid culture systems, authors unravel the molecular cross-talks between TREM2+ TAMs and LY6D- epithelial cells. They show that TREM2+ TAMs secrete OSM ligand which induces the proliferation of OSMR+LY6D- epithelial cells. In addition, author show that TREM2+ TAMs differentiate from tumor-infiltrating monocytes which self-propagate in a specific epithelial niche of the tumor microenvironment. This enables the maintenance of TREM2+ TAMs overtime within the tumor. These findings are very interesting and of particular importance for the design of new immunotherapies modulating myeloid cell function within the tumor microenvironment. The study is well designed and combines (1) state of the art high dimensional technologies enabling to explore human tissues at the spatial and transcriptomic levels and (2) in vivo controlled experiments in relevant preclinical models.

Specific comments

Figure 2h and 2i

Authors should better define how they identify myeloid cells.

Does the UMAP in 2i comprise monocytes, macrophages and dendritic cells?

How are myeloid gated? Is it by using expression of Itgam/Cd11b? This should be clarified.

Authors should provide a heatmap indicating the 10 to 20 most differentially expressed genes between the clusters C1 to C7.

Authors should also annotate all the clusters so reader can identify monocytes, DCs and macrophages within the UMAP

Fig 2k

To better identify undifferentiated monocytes authors should show the violin plots for the following transcripts: CCR2 and LY6C

C5 and C6 are not identified nor described

Fig 2n

In the pseudotime trajectory, authors should comment on the position of C5 and C6

Figure 4

Author did not clearly introduce their mouse model. This should be provided in the main result section

Authors should also explain in more details the organoid system

It is unclear for the human data (Fig4m-o) whether anti-TREM2 antibody treatment leads to depletion of TREM2+ TAMs or to their reprogramming. Could authors provide this information

Reviewer #2 (Remarks to the Author): with expertise in skin cancer

In this work, Haensel et al. attempt to establish specific changes in the BCC microenvironment and understand how these changes allow the immunosuppressive environment and potentially the resistance to immune checkpoint inhibitors. The study relies on a set of beautifully executed spatial and single cell transcriptomics and proteomics experiments followed by mechanistic validation. Although the results presented here allow a substantial progress in our understanding of the BCC tumour microenvironment warranting their publication, they fall short of the initial objectives of the

authors in relation to explaining the resistance to immune checkpoint inhibitors.

Major point:

1- Accepting the presence and role of SCAMs as well as their developmental pathway in the tumour environment as presented by authors, authors did not provide any evidence of a role in resistance to ICI. The link seems to be previous publications on TREM2+ cells, however, this was in a different context and did not include the full differentiation panel offered by authors such as VCAM expression. This should be acknowledged in discussion or the claim toned down.

2- It is unclear what distinguishes the upper from the lower tumour. In an early BCC nodule, is the entire tumour considered LT? Is it the differences in extracellular matrix or is it a characteristic of BCC cells?

3- Excitingly they demonstrate in murine tumour transplantations that the BCC growth is largely dependent on stroma. Mixing with stroma bolstered tumour growth about ten times. However Trem2 depletion only reduced tumour growth 2 fold. It should be acknowledged that other components must be important in driving the impact of the stroma on BCC growth. Most importantly it is unclear what the consequences of Trem 2 depletion are. Is anti-TREM2 reducing the number of macrophages? Is there any consequence on tumour environment? This part would benefit from some in vivo (mouse) tumour analysis. A simple IF could show if depletion of TREM2+ cells has affected the proliferation of BCC cells, the redistribution of blood vessels and tumour nodules and within the latter the Ly6D population.

Minor points:

1-Many of the patient samples obtained for analysis (3 samples in figure 1 CODEX, 13 in subsequent experiment) have not been fully characterised. It is not clear in the text whether these are advanced and inoperable samples or simply morpheiform subtypes of BCC as figured in 1. It would be interesting to see if superficial BCCs already have the undifferentiated TREM2+ cells?

2-Using additional flow cytometry cyTOF on 13 human BCCs, authors conclude the overrepresentation of a DR+ TREM2+ myeloid population within the infiltrate. The 13 samples are all BCCs and it would be interesting to show how this differs from normal skin.

3-A differentiation pathway is presented from upper to lower tumour with the mature stage having increased level of VCAM1 expression. Are cells from UT and LT both VCAM1+? In scATACseq are VCAM1 loci already open at early stages in UT? Some immunostaining images could easily resolve this question.

4- The use of BCC derived organoids is a fantastic addition to the validation methods demonstrated in this work. Given the novelty of this new tool, it would be nice to show that in vitro observations can be recapitulated in vivo upon transplantation of the organoids in NSG mice.

5- Identification of the role of OSM emanates from the addition of ligands to organoid culture. Although this is of interest a KO or inhibition strategy would be much more convincing. To this end the use of JAK inhibitors is adequate. However, current experiment presented in Fig 5o is not properly designed to address this question. Ideally there should be four conditions to compare: organoid alone, organoid + SMACs, organoid + Rux, Organoid + SMACs +Rux. It is very plausible that Ruxolitinib could have direct inhibitory effects of BCC cells in the absence of any SMAC or OSM and this should be controlled for.

6- The final figure is also a fantastic experiment to show the maintenance of SCAMs and their self renewal in tumours. However the GFP engraftment experiment seems to indicate a dilution of the adoptively transferred population between days 5 and 7 and one could wonder if at later time points as the tumour is bigger this population would get further diluted. I think authors cannot exclude completely that a subset of the SMACs continues to be derived from circulating BM derived cells.

Reviewer #3 (Remarks to the Author): with expertise in multiplexed imaging

In "Tumors assemble a spatially organized self-propagating Trem2+ myeloid niche" (NCOMMS-22-43704-T) Haensel et al. present data to support the existence of a TREM2+ myeloid population that is self-propagating in basal cell carcinoma (BCC). The single cell RNA sequencing analysis and mouse transplantation experiments provide interesting results, however, the strong conclusions about spatial organization of tumour and immune components is based on a very small number of human samples.

A few key points about the tissue imaging; about numbers of samples used and quantification framework:

1. The choice of antibodies for the CODEX imaging does not seem fit to purpose for a study that is geared toward the analysis of myeloid cells (e.g. common markers such as CD68, CD163, CD206, and Pu.1 are not used) and the analysis of immunosuppression (PD-L1 expression in myeloid cells is not assessed and Granzyme B, Perforin, Tim3, and Lag3 on T cells is not studied, etc.). Many of the antibodies that were used were not very informative; much of the analysis ignores the majority of the markers and focuses on a few markers such as CD3, FoxP3 and HLA-DR. A deeper analysis of immune differences in the existing data might have been warranted ... i.e. are other markers different in the areas that are labeled IS, LT, UT – e.g., CD8 and CD4 cells, macrophages, CD8/PD1+ cells, dendritic cells, etc. Critically, however, the multiplexed tissue analysis would have best been used to investigate the findings derived from the scRNA-sequencing data (with a more tailored and appropriate antibody selection), and not as a starting point for the study. For example, multiplexed imaging would have been optimally used to study the expression in human BCC of TREM2 relative to a variety of macrophage markers (CD68, CD163, CD206, Arg1, Pu.1), the expression of VCAM and Ki67 in those cells, and the expression of phospho-STAT3, among other markers.

2. The multiplexed tissue analysis is performed on three basal cell carcinoma (BCC) specimens (histologic types not provide; e.g. superficial, nodular, infiltrating). BCC are very common tumours so it seems that the conclusions would be based on analysis of a substantially larger cohort of samples.

3. The statements about the spatial organization require a more structured and quantitative analytical framework. The abstract and the manuscript discuss three microenvironment neighbourhoods as a key feature of BCC (UT, LT, and IS), however, these global regions/neighbourhoods are based on what appear to be arbitrary designations of the zones (how thick is UT, how thick is LT?). It is unclear why certain immediately neighbouring regions in the images of tumours #1, #2, and #3 were not include in the lassoed regions designated UT or LT or IS. Because spatial conclusions are a key feature of the study, quantitative criteria are needed to define these regions so that others can implement metrics to reproducibly make these regional assignments.

4. Clustering of markers is a helpful tool for understanding the underlying data, but the information should be described as the 'cell types' that are being captured by those clusters. Cluster 11 (green) for instance is difficult to understand because it has cells that are high in both CD4 and CD8, which should of course be mutually exclusive (this is concerning). Perhaps more clusters are needed (it would help to have an explanation for why 16 clusters were selected), perhaps there are problems with segmentation, or antibody specificity. A cell type driven analysis (with a dendrogram used to define cell types and markers) would help resolve this. Such cell type definitions would help the reader understand for instance how fibroblasts were defined and identified in Figure 1g, and what fraction of cells in the image are not being assigned a cell type because markers are not available for them in the panel (did the cell type calling capture all cells? were fibroblast defined by lack of any marker expression, i.e. defined by exclusion). Also, a cell type dictionary would help the reader understand why CD3 and FOXP3 are being used to define Tregs when that cell type would customarily be labeled using CD4 as a marker which is available in the antibody panel used (e.g. either CD3+ CD4+ FoxP3+, or CD4+ FoxP3+), and which cells are expressing HLA-DR (tumour, immune, both, which types)?

5. A small field of view is shown of what was presumably whole slide imaging of RNA in situ hybridization of Ly6D. It would help to have RNA signal detected using a spot detection algorithm and quantified and then spatial analysis performed on all LY6D+ and LY6D- tumour cells relative to HLA-DR+ immune cells and Tregs across large areas of multiple tumour samples (spatial correlation is one way to provide a quantitative measure of the relationships rather than representative fields of view) to support the conclusion that Ly6D- tumour cells are enriched for interactions with HLA-DR+ cells and not CD3 T cells. It would be useful to know if the areas of the BCC that are superficially invasive or intermediate in invasive depth might have more Ly6D expression because those regions are reported

to be less proliferative. Moreover, because there are proliferative vs. non-proliferative Ly6D⁻ cells next to each other in the same clusters of tumours (even in the LT areas), conclusions about interactions of immune cell types with each cell state (proliferative vs. non-proliferative) should be made using a spatial correlation analysis isolating proliferating from non-proliferating cells and measuring their spatial relationship with HLA-DR and CD3. This should ideally be done for each CD3 population (CD4 and CD8 and Treg). Whole slide analysis would help the reader understand the fraction of BCC epithelium that is LY6D⁺ vs -. Because these experiments rely on RNAScope, it is important to show that the CD3 and FoxP3 antibody staining worked well enough to permit interpretation of those markers, because the proteinase K treatment needed to perform RNA ish can often damage tissue antigens and make IF images difficult to interpret (often eliminating signal for certain epitopes entirely or greatly compromising signal).

6. In the example image shown in Fig 1b, there is substantial distortion in the region of the epithelium (looks out of focus or folded). It would be helpful for the authors to describe how image artifacts were accounted for (removed) in the data analysis.

7. It would help if the orientation of the tissue in Figure 1b and all subsequent representations (in Extended Data Figure 1) were the same (IF, mapping of clusters) ... side-by-side H&E, IF, and cell cluster mapping would be very useful. In Extended Data Figure 1 it is unexpected to see a sharp demarcation of tumour epithelial clusters (cluster 2, blue, top of image vs. cluster 1, red, bottom of image). Is this potentially due to uneven illumination, antigen retrieval, staining, etc.

8. Which human BCC sample is shown in Figure 2p; it would help to see the proliferation gradient relative to the RNA ish. Similar to the CODEX analysis in Figure 1, the conclusions about spatial organization require a quantitative analysis of a larger number of specimens (it is unclear if one or more were used in the analysis in this figure). In addition, the conclusion about colocalization of TREM2⁺ CD68 cells with highly proliferative LY6D⁻ cells is not quantified ... a regional assessment is shown but not a correlation between proliferating LY6D⁻ tumour cells and TREM2 CD68⁺ vs. non-proliferating LY6D⁻ tumour cells and TREM2⁺ CD68 cells.

9. With regards to the identification of VCAM and TREM2 in the stromal cells of a mouse BCC sample by RNA ish (Fig 3g), was this colocalization seen in human BCC and did that differ in tissue resident myeloid cells in samples of normal or wounded skin.

10. In the experiments transplanting male tumour cells into NOD SCID mice – more than half of the CK negative cells that show Uty signal are also F4/80 negative. Because of the diffuse processes of macrophages it is difficult to discern if the Uty signal arises from the nucleus of a F4/80⁺ cell or a neighbouring F4/80⁻ cell. Have the authors done additional phenotyping to assess those cells and also quantified the number of Uty + myeloid cells. Of the myeloid cells imaged, how many are Uty+ ... from the experiment using CD45.2 alleles, presumably this would be about half?

Regarding terminology:

1. The definition of proliferation 'gradient' is not provided – can this be shown quantitatively. The image from tumour #1 seems to show that there is high proliferation just beneath the overlying epithelium, followed by somewhat of a drop, and then an increase at the deepest areas of tumour – while there looks to be enrichment at the invasive front in this sample it does not necessarily look wholly graded. One way to show gradation would be to plot the proliferative index as a variable of the absolute distance from the epithelial-dermal junction.

2. 'Lower tumor' is likely best described as 'deep invasive' or 'deep invasive front'; 'Upper tumor' as 'superficial tumor'. In the literature, 'Immune swarm' is generally referred to as the 'tumor microenvironment' or the 'tumor immune microenvironment'. It is unclear what a term like 'immune swarm' adds, and why would a vascular endothelial marker be included as a one of three markers defining the 'immune swarm.' In Figure 1L how are individual swarms being quantified – are these

distinct pockets of variable size or circular collections of immune cells? What is the size of a swarm (i.e. upper and lower limits in terms of number of cells – so that others can quantify their own data in a reproducible manner – are they CD8 rich, CD4 rich, B cell rich, is a blood vessel needed hence the use of CD31, are they TLS-like). In Figure 1m - what regions are being analyzed for tumour-immune interactions (the entire invasive tumour, just ROIs in Fig 1i, all three tumours). Having a more comprehensive interaction mapping would be helpful – accounting for CD8, CD4, and more cell populations (more than CD3 and HLA-DR). Also, how is HLA-DR signal excluded that may be coming from the tumour cells themselves?

3. It appears to be more appropriate to refer to SCAMs (skin cancer associated macrophages) as BCCAM (basal cell carcinoma associated macrophages) because the study does not assess other skin cancer such as squamous cell carcinoma, melanoma, Merkel cell carcinoma, among others.

4. Similarly, being that the study is focused on Basal Cell Carcinoma, it seems that the title of the manuscript would preferably be more specific than “Tumors assemble a spatially organized self-propagating Trem2+ myeloid niche ...”. Rather “Basal Cell Carcinomas assemble ...”

More detail would be helpful in many of the figure legends.

Extended Data Figure 2a ... what are the green and red colors indicating.

Extended Data Figure 2b ... it is not clear from the legend or the methods what samples, what cohort and what size, what tumour types, how was high and low Trem2 expression defined for the outcome analysis (overall survival and disease-free survival. (The figure legend reads: “b. Overall survival and disease-free survival for tumours with high and low expression of Trem2”). In addition, the correlation with patient outcome highlighted is rather modest but that is not the impression given in the main text. Extended Data Figure 2c also needs more detail in order to interpret.

Minor:

Extended Data Figure 1 plot is difficult to read.

The image for CD19 in Extended Data Figure 1 shows a very high level of background (how was that accounted for).

It is perhaps not worth describing differences in immune populations at difference levels of the tumour as ‘remarkable’, particularly the invasive front which is quite distinct from the tumour rich regions in most tumours including BCC (as described in other publications including PMID: 35463364)

The sentence “In contrast to the tumor epithelium containing zones we noted areas largely devoid of tumor epithelium...” is confusing. There are clearly in every carcinoma, tumour areas and non-tumour areas.

An aim of the single cell RNAseq analysis is stated as to deconvolve the cellular make up of HLA-DR cells in the LT, but the data acquired in this analysis contains a large number of HLA-DR cells from the UT.

Reference 14 (Hansel et al.,) lacks journal or pre-print server information.

Reviewer #4 (Remarks to the Author): with expertise in tumor associated macrophages, scRNAseq

Authors mapped skin basal cell carcinoma (BCC) samples using single-cell and spatial approaches, including CODEX, to map the tumor microenvironment. They identified TREM2+ VCAM1+ macrophages within the highly proliferative neighborhood of naïve human and mouse BCCs. They further show that these macrophages promote LY6D- tumor epithelial proliferation via secretion of the ligand oncostatin-M. Interestingly they show that TREM2+ VCAM1+ macrophages proliferate within

the TME and can be maintained within serially passaged tumors. The study is of interest and well executed. However, below are some major points that limit the reviewer support for publication: The part "Trem2+Vcam1+ skin cancer-associated myeloid cells (SCAMs) are distinct from other non-cancer-associated Trem2+ myeloid cells" is expected and do not provide real new insights in the study. It is indeed expected to see transcriptomic changes in tumor associated macrophages compared to tissue resident macrophages in steady state or in wounded skin. Also, the use of the name "tricophages" is a bit odd and is not commonly used in the field.

The part "SCAMs promote tumor growth through an OSM-OSMR-STAT3 axis " is of interest but not formally validated in vivo using genetic approached removing OSM expression in TREM2+ VCAM1+ macrophages.

The last part is quite confusing. It is not really clear how much cells self-maintain vs are recruited from circulating precursors. More direct quantitative fate mapping model of monocytes could be used to support these conclusions. Finally, on the use of used RNAScope to look for the male-specific marker Uty to differentiate between transplanted and recipient cells, how the authors excluded phagocytosis of tumor cells?

Reviewer #5 (Remarks to the Author): with expertise in epigenomics, immunology

In this study, Haensel et al reported how tumors assemble a spatially organized self-propagating Trem2+ myeloid niche. Authors exploited multiple single-cell measurements including CODEX, scRNA-seq, and CYTOF in primary human and mouse tissues to better define microenvironmental neighborhoods surrounding BCC tumor epithelia. They found that TREM+ skin cancer related macrophages enable proliferation of a specific tumor epithelia population via JAK/STAT signaling. Using mouse models to test hypotheses generated from data in human tumors, they reported that tumors drives the differentiation of skin cancer related macrophages.

This work took advantage of cutting-edge single-cell strategies and relied on mouse models to fully assess the relevance of Trem2+ myeloid niche in tumors. Single-cell RNA-seq data in different mouse models were generated to assess the relevance of Trem+ cells in tumor growth. Authors can improve the quality of their study by comparing the scRNA-seq data generated throughout their study to find similarities and differences across different in vivo models. Moreover, gene lists associated with different experimental systems can enable the community to better utilize the data generated in this study.

We thank all five reviewers for their strong and positive support for our work and for their insightful comments that should enhance its impact. We have endeavored to address all comments through additional experiments, re-wording the current manuscript, and providing clear rationales. We have supplied data/figures within this point-by-point response and have indicated what components have been incorporated into the manuscript. Please note in the case of similar requests by reviewers, we display the data in one location to reduce redundancy. All changes within the manuscript have been tracked so reviewers can monitor direct changes.

Reviewer #1:

In this study, authors explore the spatial organization of the tumor microenvironment of human basal cell carcinoma by using multiplexed imaging and unbiased high dimensional analysis. Authors uncover a specific subset of tumor associated macrophages which express TREM2 and VCAM1 and which co-localize with a highly proliferative population of tumor epithelial cells (LY6D-). By a series of elegant in vivo experiments and organoid culture systems, authors unravel the molecular cross-talks between TREM2+ TAMs and LY6D- epithelial cells. They show that TREM2+ TAMs secrete OSM ligand which induces the proliferation of OSMR+Ly6D- epithelial cells. In addition, authors show that TREM2+ TAMs differentiate from tumor-infiltrating monocytes which self-propagate in a specific epithelial niche of the tumor microenvironment. This enables the maintenance of TREM2+ TAMs overtime within the tumor. These findings are very interesting and of particular importance for the design of new immunotherapies modulating myeloid cell function within the tumor microenvironment. The study is well designed and combines (1) state of the art high dimensional technologies enabling to explore human tissues at the spatial and transcriptomic levels and (2) in vivo controlled experiments in relevant preclinical models.

We thank Reviewer #1 for their positive and kind words regarding our manuscript and certainly agree that our work furthers our understanding of TREM2⁺ cells within the tumor microenvironment. Their comments certainly add to the overall storyline in a positive way.

Comment #1-1: Figure 2h and 2i. Authors should better define how they identify myeloid cells. Does the UMAP in 2i comprise monocytes, macrophages and dendritic cells? How are myeloid gated? Is it by using expression of *Itgam*/Cd11b? This should be clarified. Authors should provide a heatmap indicating the 10 to 20 most differentially expressed genes between the clusters C1 to C7. Authors should also annotate all the clusters so reader can identify monocytes, DCs and macrophages within the UMAP.

Response #1-1: We thank the reviewer for this point, and we agree that our work could benefit from additional descriptive analysis and classification of our myeloid fractions. We have added the additional analysis to better define these populations. In short, we now better distinguish between the various myeloid fractions and show how we drill down on the more mature *Trem2*⁺ myeloid fraction for subsequent analysis from our total Cd45⁺ cells. We used a combination of markers such as *Itgam* (Cd11b), *Cd68*, *Itgax* (Cd11c), and *Trem2* to define our mature myeloid fraction and markers like *Cd14* (and now *Ccr2* and *Ly6c* – see **Comment #1-2/Response #1-2**) to identify the more undifferentiated monocyte populations. Within the tumor microenvironment, there certainly are populations of cells that exhibit more macrophage-associated (*Itgam*^{Hi}), dendritic-associated (*Itgax*^{Hi}), and tissue-resident dendritic-associated (Langerhans Cells: *Cd207*^{Hi}) genes but the clusters are largely not separated from one another likely driven by regionally specific signaling within the tumor microenvironment. As such, we decided to change our annotations of key markers associated with various myeloid cell types to differentiate the clusters. The inclusion of the heatmap with the top 10 most differentially expressed genes as suggested by the reviewer further shows that there are clear gene expression differences between the different clusters. We have added the data associated with Reviewer Figure 1 to Figure 4 and Extended Data Figure 3 in the manuscript. Furthermore, we have changed the cluster identifiers for the various UMAPs in the manuscript.

Reviewer Figure 1: (A) UMAPs of different subsets from 'Mouse Total CD45+ subset', to 'Mouse total myeloid subset', to 'Mouse mature myeloid subset'. Circled clusters indicate clusters that are used in subset. (B) Heatmap of the top 10 differentially expressed genes from the 'Mouse mature myeloid subset'.

Comment #1-2/Comment #1-3: Fig 2k. To better identify undifferentiated monocytes authors should show the violin plots for the following transcripts: CCR2 and LY6C C5 and C6 are not identified nor described. Fig 2n. In the pseudotime trajectory, authors should comment on the position of C5 and C6.

Response #1-2/Response #1-3: We thank the reviewer for these comments as they enhance our interpretation of the data. We think it might be useful to merge these comments from the Reviewer to better address the points as there is substantial overlap with regards to clusters discussed. To address the point regarding monocytes, we show the Feature plots for *Cd14*, *Ccr2*, and *Ly6c1* for both the 'Mouse total myeloid subset' and the 'Mouse mature myeloid subset'. Furthermore, we have added *Ccr2* and *Ly6c* to our stacked violin plot for the 'Mouse mature myeloid subset'. We can detect *Cd14*, *Ccr2*, and low (but specific) expression of *Ly6c1* within the monocyte clusters within the 'Mouse total myeloid subset'. Within the 'Mouse mature myeloid subset', we detect *Cd14* throughout the different clusters, some *Ccr2* in specific clusters, and no *Ly6c1*, together suggesting that our 'Mouse mature myeloid subset' does not contain undifferentiated monocytes, allowing us to examine the more subtle differences between mature myeloid subsets. Based on **Comment #1-3** from the reviewer, we believe the reviewer is pointing to clusters C4 (purple) and C5 (orange) rather than C6 (pink) given that they are localized in an interesting position along the pseudotime trajectory. As such, we will describe C4-

C7 (albeit with the cluster's new names) to ensure all clusters have been addressed within the manuscript. Based on the markers and position of the purple/ $C4/Trem2^{Low}Itgam^{Hi}Ccr2^{Hi}$ and the orange/ $C5/Trem2^{Low}Itgax^{Hi}Ccr2^{Hi}$, our prediction is that these are likely more recently recruited myeloid fractions, with the purple/ $C4/Trem2^{Low}Itgam^{Hi}Ccr2^{Hi}$ being more macrophage-specific and the orange/ $C5/Trem2^{Low}Itgax^{Hi}Ccr2^{Hi}$ being more dendritic-specific. The markers and locations of the pink/ $C6/Trem2^{Hi}Itgam^{Hi}Cd14^{Hi}$ and lavender/ $C7/Trem2^{Hi}Itgax^{Hi}$ populations suggest to us that they are slightly more mature versions of their red/ $C1/Trem2^{Hi}Itgam^{Hi}Cd14^{Hi}$ and blue/ $C2/Trem2^{Hi}Itgax^{Hi}$ counterparts. We have added this new data found in Reviewer Figure 2 to Figure 4 of the manuscript.

Reviewer Figure 2: (A) Feature plots for monocyte specific markers *Cd14*, *Ccr2*, and *Ly6c1* within the Mouse total myeloid subset. (B) Feature plots for the monocyte specific markers *Cd14*, *Ccr2*, and *Ly6c1* within the Mouse mature myeloid subset. (C) UMAP of the Mouse mature myeloid subset with updated cluster labels. (D) Stacked violin plot with added *Ccr2* and *Ly6c1* gene expression markers.

Comment #1-4: Figure 4. Author did not clearly introduce their mouse model. This should be provided in the main result section.

Response #1-4: We apologize for this oversight and have now expanded our description with specific emphasis on depicting differences between our primary immunocompetent mouse model and our allograft system. In short, our primary mouse tumors are generated from *Ptch1*^{+/-};*p53*^{ff};*K14Cre-ER*;*RFP*^{f-s-f}, which after tamoxifen injections to delete *p53* from the K14-expressing skin epithelia, are irradiated using an X-ray irradiator. The irradiation leads to the loss of the second copy of *Ptch1* leading to the generation of clinically relevant tumors approximately 6 months later. Furthermore, the inclusion of *RFP*^{f-s-f} allows us to trace and separate the epithelial cells from the stromal cells. These primary or P0 tumors are allografted into Nod-Scid mice (P1 tumors) for subsequent perturbation-type studies. P0 tumors can also be allografted to make P1 tumors and can be serially allografted once more (P2 tumors) for observations on myeloid fraction. We have made this clearer in the Methods section.

Comment #1-5: Authors should also explain in more details the organoid system.

Response #1-5: We appreciate the suggestion to better explain the organoid systems. For the mouse-derived organoid model, we have now added the updated citation for our recently published paper, which was in revision at the time of submission of this manuscript (Haensel et al., 2022 Nature Communications) which fully describes the system in full detail for reference. In short, all our organoids are derived from primary mouse P0 tumors. From P0 tumor single-cell suspensions, we can either culture the bulk suspension or sort particular populations (i.e., RFP⁺ tumor epithelium, Trem2⁺ cells, etc.) and differentially combine them to address various questions. Details such as media and culture conditions are outlined in the methods. Regarding the Patient-Derived Organoid (PDO) model, our methods section lacked a citation to the original PDO paper, but we have provided it now. Furthermore, we did not have sufficient flow cytometry data showing our gating strategies for PDOs. This has been updated in Extended Data Figure 5. Details such as media and culture conditions for PDOs are summarized within the Methods section.

Comment #1-6/Comment #2-3: It is unclear for the human data (Fig4m-o) whether anti-TREM2 antibody treatment leads to depletion of TREM2⁺ TAMs or to their reprogramming. Could authors provide this information. **Comment #2-3 Part #2** from Reviewer #2: Most importantly it is unclear what the consequences of Trem2 depletion are. Is anti-Trem2 reducing the number of macrophages? Is there any consequence on tumour environment? This part would benefit from some in vivo (mouse) tumour analysis. A simple IF could show if depletion of TREM2⁺ cells has affected the proliferation of BCC cells, the redistribution of blood vessels and tumour nodules and within the latter the Ly6D population.

Response #1-6/Response #2-3 Part#2: We thank both reviewers for these similar comments and have decided to address them simultaneously. With respect to the human PDO data mentioned by Reviewer #1, flow analysis shows that we are still able to detect some TREM2⁺ cells after anti-TREM2 treatment but we find that when we examine the TREM2⁺ cells, there appears to be a reduction in TREM2 surface marker levels, as compared to the TREM2⁺ cells of anti-IgG treated tumors, in 3 out of the 4 samples. Interestingly, the sample whose TREM2⁺ cells were not affected (BCC5/Patient #1) showed little to no change in ApoGreen% indicating a correlation between antibody efficacy and tumor epithelial response. As we did not include other potential markers in our flow panel and those human samples were exhausted, we turned to our functional mouse experiments for additional insight.

Furthermore, in contrast to our mouse allograft experiments, we cannot efficiently track TREM2⁺ cells number within our PDOs as it is impossible to ensure the same number of cells across treatment groups. In our functional mouse experiments, we treated allografted tumors (P1) with anti-Trem2 antibodies to interrogate its effect on BCC tumor growth and SCAMs. We found reduced tumor growth capacity relative to anti-IgG controls (now with n = 5). In this model, it is critical to point out that there would not be additional recruitment of CD45.2⁺Trem2⁺ SCAMs. CD45.1⁺Trem2⁺ cells could in principle be recruited as they are found within allografted tumors, but from our scRNA-Seq analysis, this myeloid fraction has reduced *Trem2*, *Osm*, and *Vcam1* expression suggesting a more defective macrophage from the immunocompromised recipient mouse. SCAMs (CD45.2⁺Trem2⁺) can maintain themselves within the tumor due to their self-propagating behavior. Using RNAScope, we stained for *Trem2* and *Vcam1* in tumors from mice that were treated with anti-IgG and anti-Trem2. We made two key observations. First, we found that *Trem2* was reduced in the anti-Trem2 treated mice. Interestingly, the levels of *Vcam1* were relatively unchanged. This reduction in *Trem2* levels would suggest myeloid reprogramming. Second, we found that the amount of *Trem2*⁺ and/or *Vcam1*⁺ cells were reduced in the anti-Trem2 treated mouse tumors suggesting some depletion of the cells. Overall, it appears that both reprogramming and depletion of SCAMs might be consequences of anti-Trem2 treatment. What does appear clear though is that anti-Trem2 treatment leads to reduced proliferative capacity of the tumor epithelium (by Ki67 levels), which is consistent with the observed reduction in tumor growth. Subsequent studies outside the scope of this work in immunocompetent hosts would be needed to better define the gene expression changes of anti-Trem2 treatment within a host, which could in principle recruit functional myeloid cells. We have added this new data found in Reviewer Figure 3 to Figure 6 and Extended Data Figure 5.

Reviewer Figure 3: (A) ApoGreen% of EPCAM⁺ epithelial cells from PDOs treated with anti-IgG or anti-TREM2. (B) Flow histograms showing the intensity of TREM2 in TREM2⁺ cells from PDOs treated with anti-IgG or anti-TREM2. (C) Diagram of tumor formation and antibody treatment strategies. (D) Updated tumor sizes after treatment with anti-IgG and anti-Trem2 with the additional sample labeled in red, which was used for subsequent experiments (n-value now = 5). (E) RNAScope analysis of tumors treated with anti-IgG or anti-Trem2, which were stained for *Trem2* and *Vcam1*. Cells are indicated with yellow outlines. Punchouts of *Trem2* signal from cells are shown on the right. (F) Plot showing the various intensities of *Trem2/Vcam1* in the anti-IgG and anti-Trem2 treated mice. (G) The average number of *Trem2*⁺ and/or *Vcam1*⁺ cells per frame in the anti-IgG or anti-Trem2 treated mice (n = 2 tumors with 3 frames per tumor). (H) Ki67 immunofluorescence with subsequent quantification of anti-IgG and anti-Trem2 treated mice (n = 2 tumors with 3 frames per tumor).

Reviewer #2:

In this work, Haensel et al. attempt to establish specific changes in the BCC microenvironment and understand how these changes allow the immunosuppressive environment and potentially the resistance to immune checkpoint inhibitors. The study relies on a set of beautifully executed spatial and single cell transcriptomics and proteomics experiments followed by mechanistic validation. Although the results presented here allow a substantial progress in our understanding of the BCC tumour microenvironment warranting their publication, they fall short of the initial objectives of the authors in relation to explaining the resistance to immune checkpoint inhibitors.

We thank Reviewer #2 for their positive and kind words regarding our various experiments coupling spatial, single-cell-based, and functional experiments to elucidate the function of Trem2⁺ cells. We believe that the inclusion of more data as well as some changes in wording will address their main concern regarding immune checkpoint inhibitors. Our goal was to describe the novel functions of Trem2⁺ cells while acknowledging their previously identified function in cancer immunotherapy.

Comment #2-1: Accepting the presence and role of SCAMs as well as their developmental pathway in the tumour environment as presented by authors, authors did not provide any evidence of a role in resistance to ICI. The link seems to be previous publications on TEM2⁺ cells, however, this was in a different context and did not include the full differentiation panel offered by authors such as VCAM expression. This should be acknowledged in discussion or the claim toned down.

Response #2-1: We thank the reviewer for this comment. We agree that as the article currently stands, we do not necessarily draw connections directly to immune checkpoint blockade (ICB) therapy. As the reviewer notes, we do certainly draw on literature that does suggest a causal role for the presence of TREM2⁺ cells and resistance to ICB therapy. Our intention was to build on some of these findings but to propose an immunosuppression-independent role for TREM2⁺ cells by showing that TREM2⁺ cells can directly promote tumor growth via OSM. However, we would certainly agree that putting TREM2⁺ cells in the context of ICB would be greatly beneficial to this study. We now include functional data from our patient-derived organoid (PDO) system that includes anti-PD1 treatments with and without anti-TREM2 treatments. Our data suggest that in 3/4 cases, the addition of anti-TREM2 antibodies promotes greater cell death of anti-PD1 treated epithelial cells, in agreement with previous literature. Although we acknowledge that this may not directly address the role of TREM2⁺ cells in tumor resistance to ICB, we feel that this certainly would enhance the case. We have added this new data found in Reviewer Figure 4 to Figure 6 of the manuscript.

Reviewer Figure 4: (A) Diagram of PDO organoid system with different antibody treatments. (B) Representative flow diagrams showing ApoGreen% after treatment with different antibodies. (C) Summary of all treatments (algG, aPD1, aTREM2, aPD1aTREM2) across four different patient tumor samples.

Comment #2-2: It is unclear what distinguishes the upper from the lower tumour. In an early BCC nodule, is the entire tumour considered LT? Is it the differences in extracellular matrix or is it a characteristic of BCC cells?

Response #2-2: We thank the reviewer for this comment, which appears shared by Reviewer #3 in some general sense. What is useful about BCCs is that you can capture the patient's normal epidermis as a key landmark to better understand the spatial z-axis, allowing you to distinguish between the upper and lower parts of the tumor. In general, there are four key concepts that distinguish the UT and LT:

1. There are biological states – specifically the enhanced proliferative nature of the tumor epithelium within the LT. We have quantified this computationally from our CODEX analysis as well as added additional tumors ($n = 2$), which we stained to further confirm our statistically significant CODEX results.
2. There are distinct epithelial states – specifically those that are $LY6D^+$ and $LY6D^-$, which have previously been defined as being resistant to canonical hedgehog inhibitors. We find that there is more $LY6D$ within the UT region, which logically makes sense given the fact that we find more SCAMs, which interact with $Ly6d^+$ tumor epithelial cells, within the LT region. We show this through additional staining of larger regions.
3. There are distinct cellular makeups within the UT and LT – specifically the enrichment of $HLA-DR^+$ and $CD3^+FOXP3^+$ cells within the LT region. We show this by our existing CODEX analysis.
4. Finally, there are distinct differentiation/polarization states of macrophages that span the UT and LT. Using a combination of new scRNA-Seq (now including 2 additional samples), Monocle analysis, and new staining data, we have made this point clearer. We have also included a diagram that better summarizes these key points.

We have added this new data found in Reviewer Figure 5 to Figure 1, Figure 2, and Figure 3 within the manuscript.

Reviewer Figure 5: (A) Additional protein staining of Ki67 confirms statistically significant spatial proliferation results (n = 2). (B) RNAScope of *LY6D* levels throughout the tumor. Quantification of *LY6D*⁺ clone size in the UT and LT regions. (C) Merged analysis (n = 4 patients) of *HLA-DR*⁺. (D) Feature plots showing expression of *CD68* and *TREM2* from the merged analysis. (E) Subset of *HLA-DR*⁺ cells from an individual patient. (F) The subset of the *CD68*⁺ macrophage population with key markers labeled. (G) Feature plots showing expression of *CD68* and *TREM2*. (H) Monocle plot showing the predicted trajectory towards an *ITGAM*⁺*CD68*⁺*TREM2*⁺ state. (I) RNAScope of *CD68* and *TREM2* throughout the

different regions of the tumor. Punchouts of the TREM2 signal in the UT and LT regions. (J) Quantification of the TREM2 signal of the CD68⁺ cells in different regions of the tumor. (K) Polarization summary that spans the UT to LT.

Comment #2-3 Part#1: Excitingly they demonstrate in murine tumour transplantations that the BCC growth is largely dependent on stroma. Mixing with stroma bolstered tumour growth about ten times. However Trem2 depletion only reduced tumour growth 2 fold. It should be acknowledged that other components must be important in driving the impact of the stroma on BCC growth.

Response #2-3 Part#1: This is a good point raised by the reviewer. We certainly acknowledge that there are likely other cellular players within the tumor stroma that would support tumor growth and include these sentiments in the text.

Comment #2-3 Part#2: Most importantly it is unclear what the consequences of Trem2 depletion are. Is anti-Trem2 reducing the number of macrophages? Is there any consequence on tumour environment? This part would benefit from some in vivo (mouse) tumour analysis. A simple IF could show if depletion of TREM2⁺ cells has affected the proliferation of BCC cells, the redistribution of blood vessels and tumour nodules and within the latter the Ly6D population.

Response #2-3 Part#1: This comment is addressed in **Response #1-6** as Reviewer #1 had a similar comment.

Minor Comment #2-1: Many of the patient samples obtained for analysis (3 samples in figure 1 CODEX, 13 in subsequent experiment) have not been fully characterised. It is not clear in the text whether these are advanced and inoperable samples or simply morpheiform subtypes of BCC as figured in 1. It would be interesting to see if superficial BCCs already have the undifferentiated TREM2⁺ cells?

Minor Response #2-1: As part of our human subject's agreement, we obtain our samples from our MOH's surgery unit without identifiers including clinical history; therefore, while we know these tumors are drug-naïve, we don't know whether they went on to become multiply resistant or inoperable. In early lesions, we can see TREM2⁺ myeloid cells, consistent with the notion that the tumor epithelium drives the self-propagating tumor-immune niche. We have however included all H/E now in Extended Data Figure 2.

Minor Comment #2-2: Using additional flow cytometry cyTOF n 13 human BCCs, authors conclude the overrepresentation of a DR⁺ TREM2⁺ myeloid population within the infiltrate. The 13 samples are all BCCs and it would be interesting to show how this differs from normal skin.

Minor Response #2-2: We refer the reviewer to Figure 5 of this manuscript, which extensively analyzes the key differences between myeloid populations between normal, wounded, and tumor.

Minor Comment #2-3: A differentiation pathway is presented from upper to lower tumor with the mature stage having increased level of VCAM1 expression. Are cells from UT and LT both VCAM1⁺? In scATACseq are VCAM1 loci already open at early stages in UT? Some immunostaining images could easily resolve this question.

Minor Response #2-3: We thank the reviewer for this insightful query. We have addressed several of these ideas/concepts in Reviewer #2 Comment 2-2, where we better show the polarization gradient through the use of Monocle analysis from new human scRNA-Seq data. As we had previously suggested, the UT contained cells that are CD68⁺TREM2⁻ whereas the LT contained cells that were CD68⁺TREM2⁺. We now show that there are cells that are CD68⁺TREM2⁺VCAM1⁺ by scRNA-Seq as

well. The Monocle analysis would suggest that this $CD68^+TREM2^+VCAM1^+$ state represents the most mature and would likely be within the LT and not the UT region. We have added this new data found in Reviewer Figure 6 to Figure 5 of the manuscript.

Reviewer Figure 6: (A) Subset of $HLA-DR^+$ cells from an individual patient. (B) The subset of the $CD68^+$ macrophage population with key markers labeled. (C) Feature plots showing expression of $CD68$ and $TREM2$. (D) Monocle plot showing the predicted trajectory towards an $ITGAM^+CD68^+TREM2^+$ state. (E) Feature plot showing the enrichment of $VCAM1$ within the blue terminal cluster. (F) Updated polarization summary.

Minor Comment #2-4: The use of BCC derived organoids is a fantastic addition to the validation methods demonstrated in this work. Given the novelty of this new tool, it would be nice to show that in vitro observations can be recapitulated in vivo upon transplantation of the organoids in NSG mice.

Minor Response #2-4: We thank the reviewer for their comments on our organoid system. This organoid system has recently been published by our group: Haensel et al., 2022 Nature Communications. This manuscript was in the final stages of revision at the time of submission of this current manuscript. This organoid tool has been previously used and cited. Unfortunately, due to the extensive experimentation optimization needed to transplant organoids into Nod Scid mice, we believe this experiment to be outside the scope of this manuscript and not feasible within the resubmission timeline.

Minor Comment #2-5: Identification of the role of OSM emanates from the addition of ligands to organoid culture. Although this is of interest a KO or inhibition strategy would be much more convincing. To this end the use of JAK inhibitors is adequate. However, current experiment presented in Fig 5o is not properly designed to address this question. Ideally there should be four conditions to compare: organoid alone, organoid + SMACs, organoid + Rux, Organoid + SMACs +Rux. It is very plausible that Ruxolitinib could have direct inhibitory effects of BCC cells in the absence of any SMAC or OSM and this should be controlled for.

Minor Response #2-5: To enhance the notion that the ligand Osm promotes tumor growth, we utilized an in vivo neutralization strategy with anti-Osm antibodies to show that targeting this ligand reduces tumor growth. This in vivo approach using three different primary tumors provides strong additional support for the central role of Osm in the tumor immune niche. We have added this new data found in Reviewer Figure 7 to Figure 7 of the manuscript.

Reviewer Figure 7: (A) Diagram of in vivo Osm-depletion strategy after initial tumor formation. (B) Tumor growth curve of mice treated with either algG or aOsm antibodies while monitoring tumor growth.

Minor Comment #2-6: The final figure is also a fantastic experiment to show the maintenance of SCAMs and their self renewal in tumours. However the GFP engraftment experiment seems to indicate a dilution of the adoptively transferred population between days 5 and 7 and one could wonder if at later time points as the tumour is bigger this population would get further diluted. I think authors cannot exclude completely that a subset of the SMACs continues to be derived from circulating BM derived cells.

Minor Response #2-6: The reviewer is spot on in their critique, a view we share that perhaps wasn't fully translated in our writing of the manuscript. This experiment aimed to simply show that an undifferentiated population of monocytes can quickly mature within the tumor microenvironment and not get at questions related to macrophage kinetics because as the reviewer points out, recruitment from the bone marrow would likely occur.

Reviewer #3:

In “Tumors assemble a spatially organized self-propagating Trem2+ myeloid niche” (NCOMMS-22-43704-T) Haensel et al. present data to support the existence of a TREM2+ myeloid population that is self-propagating in basal cell carcinoma (BCC). The single cell RNA sequencing analysis and mouse transplantation experiments provide interesting results, however, the strong conclusions about spatial organization of tumour and immune components is based on a very small number of human samples.

We thank the reviewer for their positive comments. We feel that the main novelty of this work is the self-propagating tumor-immune niche involving LY6D⁻ tumor epithelial cells with TREM2⁺ myeloid cells seen in all samples tested, not the spatial organization of the tumor. The starting point of the study derives from our recently published analysis of the different keratocarcinoma epithelial tumor subtypes to better understand what general tumor types are associated with them (Haensel et al., 2022, Nature Communications). We discover a new myeloid phenotype in Fig. 1 and focus on its properties in subsequent figures, with additional analysis and tools to be elaborated in subsequent publications. Our intention from this work was to use spatial-based tools to drive initial observations and frame the experimental system in an unbiased manner with our substantive claims based on some statistical inference but more focused on rigorous functional and perturbation-based experiments.

A few key points about the tissue imaging; about numbers of samples used and quantification framework:

Comment #3-1: The choice of antibodies for the CODEX imaging does not seem fit to purpose for a study that is geared toward the analysis of myeloid cells (e.g. common markers such as CD68, CD163, CD206, and Pu.1 are not used) and the analysis of immunosuppression (PD-L1 expression in myeloid cells is not assessed and Granzyme B, Perforin, Tim3, and Lag3 on T cells is not studied, etc.). Many of the antibodies that were used were not very informative; much of the analysis ignores the majority of the markers and focuses on a few markers such as CD3, FoxP3 and HLA-DR. A deeper analysis of immune differences in the existing data might have been warranted ... i.e. are other markers different in the areas that are labeled IS, LT, UT – e.g., CD8 and CD4 cells, macrophages, CD8/PD1+ cells, dendritic cells, etc. Critically, however, the multiplexed tissue analysis would have best been used to investigate the findings derived from the scRNA-sequencing data (with a more tailored and appropriate antibody selection), and not as a starting point for the study. For example, multiplexed imaging would have been optimally used to study the expression in human BCC of TREM2 relative to a variety of macrophage markers (CD68, CD163, CD206, Arg1, Pu.1), the expression of VCAM and Ki67 in those cells, and the expression of phospho-STAT3, among other markers.

Response #3-1: We thank the reviewer for their extensive insight which we can certainly use as the basis for future experiments. Our intention was to initiate an unbiased analysis of stromal components and the general architecture of BCCs for the first time using an existing panel of antibodies. Subsequent work in our lab to be published under separate cover on this project is now focused on the design of more informative panels (based on findings in this work) and integration algorithms that will inform the greater spatial resolution of neighborhood types/functions.

Comment #3-2: The multiplexed tissue analysis is performed on three basal cell carcinoma (BCC) specimens (histologic types not provide; e.g. superficial, nodular, infiltrating). BCC are very common tumours so it seems that the conclusions would be based on analysis of a substantially larger cohort of samples.

Response #3-2: We thank the reviewer for this comment. We would agree with the reviewer that a larger cohort of samples would be needed if our work was not subsequently backed up with extensive functional validations. However, given the fact that we use a combination of scRNA-Seq of primary

human (now with additional samples included to further enhance points) and mouse samples, scATAC-Seq of mouse samples, CyTOF analysis with 13 human BCC samples, extensive analysis of tumor vs. normal vs. wounded states via scRNA-Seq and bulk ATAC-Seq analysis, in vivo primary mouse tumor models, in vitro tumor organoid models, ex vivo patient-derived tumor organoid models, receptor/ligand based analysis with subsequent in vitro and now in vivo validation, as well as novel monocyte in vivo maturation assays, we believe that we have completed a sufficient amount of work to warrant publication. As indicated above, subsequent work in our lab to be published under separate cover on this project is now focused on the design of more informative panels and integration algorithms that will inform the greater spatial resolution of neighborhood types/functions as there are many other interesting and clinically relevant tumor and stromal populations to investigate in the context of BCC.

Comment #3-3: The statements about the spatial organization require a more structured and quantitative analytical framework. The abstract and the manuscript discuss three microenvironment neighbourhoods as a key feature of BCC (UT, LT, and IS), however, these global regions/neighbourhoods are based on what appear to be arbitrary designations of the zones (how thick is UT, how thick is LT?). It is unclear why certain immediately neighbouring regions in the images of tumours #1, #2, and #3 were not include in the lassoed regions designated UT or LT or IS. Because spatial conclusions are a key feature of the study, quantitative criteria are needed to define these regions so that others can implement metrics to reproducibly make these regional assignments.

Response #3-3: Please see the initial comments above. While the reviewer brings up interesting points, we respectfully remind the reviewer that the main novelty of this work is the self-propagating tumor-immune niche involving LY6D⁻ tumor epithelial cells with TREM2⁺ myeloid cells seen in all samples tested, not the detailed spatial organization of the tumor that will be published under separate cover. The starting point of the study derives from our recently published analysis of the different keratocarcinoma epithelial tumor subtypes and to better understand what general tumor types are associated with them. The reviewer is correct in the critique that we have not completed a highly intricate computational and statistical spatial analysis of the tumors. Rather, we have taken advantage of a key landmark region, the patient's normal epidermis, as a feature not seen in other spatial types of analysis. From here we make a combination of generalizable conclusions, which are seen throughout our CODEX and subsequent analysis. As described in **Response #2-2**, we have done a better job now to define four key concepts that distinguish the various UT, LT, and IS regions (biological states, epithelial states, particular cellular compositions, and differentiation/polarization states). In the end, we identify a myeloid cell of interest, which is extensively functionally validated in the subsequent figures. Future detailed analyses published elsewhere will determine whether thickness or transition zone composition affects the cellular neighborhood composition, proliferation rate, or treatment sensitivity.

Comment #3-4: Clustering of markers is a helpful tool for understanding the underlying data, but the information should be described as the 'cell types' that are being captured by those clusters. Cluster 11 (green) for instance is difficult to understand because it has cells that are high in both CD4 and CD8, which should of course be mutually exclusive (this is concerning). Perhaps more clusters are needed (it would help to have an explanation for why 16 clusters were selected), perhaps there are problems with segmentation, or antibody specificity. A cell type driven analysis (with a dendrogram used to define cell types and markers) would help resolve this. Such cell type definitions would help the reader understand for instance how fibroblasts were defined and identified in Figure 1g, and what fraction of cells in the image are not being assigned a cell type because markers are not available for them in the panel (did the cell type calling capture all cells? were fibroblast defined by lack of any marker expression, i.e. defined by exclusion). Also, a cell type dictionary would help the reader understand why CD3 and FOXP3 are being used to define Tregs when that cell type would customarily be labeled using CD4 as a marker which is available in the antibody panel used (e.g. either CD3⁺ CD4⁺ FoxP3⁺, or CD4⁺ FoxP3⁺), and which cells are expressing HLA-DR (tumour, immune, both, which types)?

Response #3-4: We thank the reviewer for this comment although we respectfully disagree with several conclusions. It is not surprising to see T cells, both CD4 and CD8 clustered together in a dataset that contains vastly different cell types (i.e., fibroblasts, epithelial cells, endothelial cells, etc.). Whether the metadata used for dimensional reduction is based on fluorescence from antibody signal or gene expression like with scRNA-Seq data, T cells of any subtype are going to much more like one another as compared to the other cell types (such as fibroblasts) that are present. We don't believe that the biased addition of clusters is the appropriate strategy in any dimensional reduction plotting tool. We do agree that additional information of what each marker identifies/is used for would be beneficial. We have added additional marker information to Extended Data Figure 1.

Comment #3-5: A small field of view is shown of what was presumably whole slide imaging of RNA in situ hybridization of Ly6D. It would help to have RNA signal detected using a spot detection algorithm and quantified and then spatial analysis performed on all LY6D⁺ and LY6D⁻ tumour cells relative to HLA-DR⁺ immune cells and Tregs across large areas of multiple tumour samples (spatial correlation is one way to provide a quantitative measure of the relationships rather than representative fields of view) to support the conclusion that Ly6D⁻ tumour cells are enriched for interactions with HLA-DR⁺ cells and not CD3 T cells. It would be useful to know if the areas of the BCC that are superficially invasive or intermediate in invasive depth might have more Ly6D expression because those regions are reported to be less proliferative. Moreover, because there are proliferative vs. non-proliferative Ly6D⁻ cells next to each other in the same clusters of tumours (even in the LT areas), conclusions about interactions of immune cell types with each cell state (proliferative vs. non-proliferative) should be made using a spatial correlation analysis isolating proliferating from non-proliferating cells and measuring their spatial relationship with HLA-DR and CD3. This should ideally be done for each CD3 population (CD4 and CD8 and Treg). Whole slide analysis would help the reader understand the fraction of BCC epithelium that is LY6D⁺ vs -. Because these experiments rely on RNAScope, it is important to show that the CD3 and FoxP3 antibody staining worked well enough to permit interpretation of those markers, because the proteinase K treatment needed to perform RNA ish can often damage tissue antigens and make IF images difficult to interpret (often eliminating signal for certain epitopes entirely or greatly compromising signal).

Response #3-5: This is a complex request by the reviewer which we will address in parts. First, we would like to clarify that no RNAScope was done in conjunction with CODEX analysis. As such there is no 'whole slide' CODEX analysis views containing *LY6D*. Images were taken on different instruments as noted in the methods. However, to bolster the analysis, we have (please see **Comment #2-2**) now included new 'whole slide' (non-CODEX) images showing the distinct distribution of *LY6D* throughout the tumor. Furthermore, we have also taken measurements to bolster the associations/statements of the proximity of HLA-DR⁺ and CD68⁺TREM2⁺ cells to *LY6D*⁻ cells. We have added this new data found in Reviewer Figure 8 to Figure 1 and Figure 3. We would also like to reiterate that this work is not focused on T cells, but on myeloid-derived cell types. Ongoing work in the lab outside this publication will address the various T cell subtypes in the context of the tumor epithelial and myeloid cell types defined in this publication. No RNAScope experiments have been done in conjunction with CD3 or FOXP3 antibody-based staining so the proposed quality control would not be needed. As noted in the methods section, for the HLA-DR antibody co-staining experiment, we use the commercially available product from ACD Bio where primary antibody staining is done before the various pre-treatments associated with RNAScope.

Reviewer Figure 8: (A) Quantitative measurements of the distance between HLA-DR⁺ cells and LY6D⁻ and LY6D⁺ cells (n = 3). (B) Quantitative measurements of the distance between CD68⁺TREM2⁺ cells and LY6D⁻ and LY6D⁺ cells (n = 3).

Comment #3-6: In the example image shown in Fig 1b, there is substantial distortion in the region of the epithelium (looks out of focus or folded). It would be helpful for the authors to describe how image artifacts were accounted for (removed) in the data analysis.

Response #3-6: We thank the reviewer for pointing this out and have clarified this within the text. In short, due to the ability to discriminate between normal epidermis and tumor, we gated out distorted regions as well as the normal epidermis within the CODEX MAV program.

Comment #3-7: It would help if the orientation of the tissue in Figure 1b and all subsequent representations (in Extended Data Figure 1) were the same (IF, mapping of clusters) ... side-by-side H&E, IF, and cell cluster mapping would be very useful. In Extended Data Figure 1 it is unexpected to see a sharp demarcation of tumour epithelial clusters (cluster 2, blue, top of image vs. cluster 1, red, bottom of image). Is this potentially due to uneven illumination, antigen retrieval, staining, etc.

Response #3-7: We have changed the orientation of the images in Extended Data Figure 1 as requested. It is not clear to us why there are demarcations noted by the reviewer in the Voronoi plot. Because it appears distracting to readers, we have removed this image as it does not add to the overall conclusions.

Comment #3-8: Which human BCC sample is shown in Figure 2p; it would help to see the proliferation gradient relative to the RNA ish. Similar to the CODEX analysis in Figure 1, the conclusions about spatial organization require a quantitative analysis of a larger number of specimens (it is unclear if one or more were used in the analysis in this figure). In addition, the conclusion about colocalization of TREM2⁺ CD68 cells with highly proliferative LY6D⁻ cells is not quantified ... a regional assessment is shown but not a correlation between proliferating LY6D⁻ tumour cells and TREM2⁺ CD68⁺ vs. non-proliferating LY6D⁻ tumour cells and TREM2⁺ CD68 cells.

Response #3-8: As indicated in **Response #3-5**, Fig. 2p is not a CODEX image. As noted in the response to **Response #2-2**, we have enhanced the points raised regarding a larger number of subsets (now n = 4 different tumors) for the points regarding the spatial maturation of macrophages (ITGAM⁺CD68⁺ to ITGAM⁺CD68⁺TREM2⁺). We have added additional analysis now showing that indeed, as clear by the images and from our previous understanding of where we see LY6D within the tumor epithelium, the CD68⁺TREM2⁺ cells are indeed closer to the LY6D⁻ cells.

Comment #3-9: With regards to the identification of VCAM and TREM2 in the stromal cells of a mouse BCC sample by RNA ish (Fig 3g), was this colocalization seen in human BCC and did that differ in tissue resident myeloid cells in samples of normal or wounded skin.

Response #3-9: This is a good question by the reviewer. We would point them to the analysis we had shown in **Comment #2-3**, which shows the new Monocle results, highlighting the existence of a $CD68^+TREM2^+VCAM1^+$ terminal state. Regarding the differences between mouse BCC tumors, normal skin, and wounded skin, we would like to point out our extensive analysis shown in Figure 5 where we directly answer these questions.

Comment #3-10: In the experiments transplanting male tumour cells into NOD SCID mice – more than half of the CK negative cells that show Uty signal are also F4/80 negative. Because of the diffuse processes of macrophages it is difficult to discern if the Uty signal arises from the nucleus of a F4/80+ cell or a neighbouring F4/80- cell. Have the authors done additional phenotyping to assess those cells and also quantified the number of Uty + myeloid cells. Of the myeloid cells imaged, how many are Uty+ ... from the experiment using CD45.2 alleles, presumably this would be about half?

Response #3-10: As pointed out in the Results section, our *Uty* analysis is strongly supported by the more rigorous and quantitative Cd45.1/Cd45.2-based analysis that clearly shows that a large portion of the Trem2⁺ cells are derived from the allograft/donor. We kept the imaging-based Uty+ assessment as orthogonal support.

Regarding terminology:

Comment #3-11: The definition of proliferation ‘gradient’ is not provided – can this be shown quantitatively. The image from tumour #1 seems to show that there is high proliferation just beneath the overlying epithelium, followed by somewhat of a drop, and then an increase at the deepest areas of tumour – while there looks to be enrichment at the invasive front in this sample it does not necessarily look wholly graded. One way to show gradation would be to plot the proliferative index as a variable of the absolute distance from the epithelial-dermal junction.

Response #3-11: We appreciate the reviewer’s comment and have removed the term “gradient” to avoid confusion.

Comment #3-12: ‘Lower tumor’ is likely best described as ‘deep invasive’ or ‘deep invasive front’; ‘Upper tumor’ as ‘superficial tumor’. In the literature, ‘Immune swarm’ is generally referred to as the ‘tumor microenvironment’ or the ‘tumor immune microenvironment’. It is unclear what a term like ‘immune swarm’ adds, and why would a vascular endothelial marker be included as a one of three markers defining the ‘immune swarm.’ In Figure 1L how are individual swarms being quantified – are these distinct pockets of variable size or circular collections of immune cells? What is the size of a swarm (i.e. upper and lower limits in terms of number of cells – so that others can quantify their own data in a reproducible manner – are they CD8 rich, CD4 rich, B cell rich, is a blood vessel needed hence the use of CD31, are they TLS-like). In Figure 1m - what regions are being analyzed for tumour-immune interactions (the entire invasive tumour, just ROIs in Fig 1i, all three tumours). Having a more comprehensive interaction mapping would be helpful – accounting for CD8, CD4, and more cell populations (more than CD3 and HLA-DR). Also, how is HLA-DR signal excluded that may be coming from the tumour cells themselves?

Response #3-12: Please see responses to this critique under **Response #3-3**. We respectfully disagree with the reviewer’s assertion that ‘Immune Swarm’ is generally referred to as ‘tumor microenvironment’, as the latter is a quite ill-defined term. We provide quantitative definitions of the terms based on previous epithelial tumor cell states (proliferation), $LY6D^-$, $LY6D^+$, or no tumor

epithelium, and the cellular arrangement surrounding these three tumor types. The immune swarm highlights a distinct immune and tumor profile not seen in the other two neighborhoods. Importantly the main thrust of the paper is to functionally validate the role of one of the key cells in the LT neighborhood. We have removed the analysis looking at the swarms of CD3⁺ cells as we do not want to generate any confusion within our work. We don't currently know why a vascular marker would help define the neighborhood, but it provides a testable hypothesis for future studies. We provide additional clarity to these thoughts in both the introduction and discussion.

Comment #3-13: It appears to be more appropriate to refer to SCAMs (skin cancer associated macrophages) as BCCAM (basal cell carcinoma associated macrophages) because the study does not assess other skin cancer such as squamous cell carcinoma, melanoma, Merkel cell carcinoma, among others.

Response #3-13: We respectfully disagree with the reviewer. Our analysis of untreated non-melanoma skin cancers demonstrates they are in a continuum of epigenetic states representing the plasticity of the tissue type, not basal or squamous or Merkel cell carcinoma. Our work and others have shown that there is similar plasticity in the tumor microenvironment, leading to the term SCAM. We anticipate there will be many epigenetic myeloid states associated with non-melanoma skin cancers and are focusing enormous effort on describing and validating their functions in our lab.

Comment #3-14: Similarly, being that the study is focused on Basal Cell Carcinoma, it seems that the title of the manuscript would preferably be more specific than "Tumors assemble a spatially organized self-propagating Trem2⁺ myeloid niche ...". Rather "Basal Cell Carcinomas assemble ..."

Response #3-14: See **Response #3-13** for our strong desire to keep the same title.

Comment #3-15: More detail would be helpful in many of the figure legends. Extended Data Figure 2a ... what are the green and red colors indicating. Extended Data Figure 2b ... it is not clear from the legend or the methods what samples, what cohort and what size, what tumour types, how was high and low Trem2 expression defined for the outcome analysis (overall survival and disease-free survival. (The figure legend reads: "b. Overall survival and disease-free survival for tumours with high and low expression of Trem2"). In addition, the correlation with patient outcome highlighted is rather modest but that is not the impression given in the main text. Extended Data Figure 2c also needs more detail in order to interpret.

Response #3-15: We thank the reviewer for their careful reading of the manuscript and have addressed these concerns within the text.

Minor Comment #3-1: Extended Data Figure 1 plot is difficult to read.

Minor Response #3-1: We have addressed this in the Extended Data Figure 1.

Minor Comment #3-2: The image for CD19 in Extended Data Figure 1 shows a very high level of background (how was that accounted for).

Minor Response #3-2: This is a good catch by the reviewer. This was supposed to be CD138, which when examining the heatmap in Extended Data Figure 1, we note that CD138 levels seem highly specific to clusters.

Minor Comment #3-3: It is perhaps not worth describing differences in immune populations at difference levels of the tumour as "remarkable", particularly the invasive front which is quite distinct from

the tumour rich regions in most tumours including BCC (as described in other publications including PMID: 35463364)

Response #3-3: We have removed the word 'remarkable'. Compared to PMID: 35463364, our manuscript has extensive functional work allowing us to investigate the function of immune cells in regulating distinct clinically relevant tumor epithelial populations.

Minor Comment #3-4: The sentence "In contrast to the tumor epithelium containing zones we noted areas largely devoid of tumor epithelium..." is confusing. There are clearly in every carcinoma, tumour areas and non-tumour areas.

Response #3-4: We have adjusted the wording.

Minor Comment #3-5: An aim of the single cell RNAseq analysis is stated as to deconvolve the cellular make up of HLA-DR cells in the LT, but the data acquired in this analysis contains a large number of HLA-DR cells from the UT.

Minor Response #3-5: This is an issue with all scRNA-Seq approaches and we absolutely acknowledge this. Therefore, we followed up with additional staining of the different clusters – *CD207*, *CD68*, and *TREM2*, etc. which we were able to attribute to locations within the tumor.

Minor Comment #3-6: Reference 14 (Hansel et al.,) lacks journal or pre-print server information.

Minor Response #3-6: Haensel et al. 2022 is now published (PMID: 36473848)

Reviewer #4:

Authors mapped skin basal cell carcinoma (BCC) samples using single-cell and spatial approaches, including CODEX, to map the tumor microenvironment. They identified TREM2⁺ VCAM1⁺ macrophages within the highly proliferative neighborhood of naïve human and mouse BCCs. They further show that these macrophages promote LY6D⁺ tumor epithelial proliferation via secretion of the ligand oncostatin-M. Interestingly they show that TREM2⁺ VCAM1⁺ macrophages proliferate within the TME and can be maintained within serially passaged tumors. The study is of interest and well executed. However, below are some major points that limit the reviewer support for publication:

We very much thank Reviewer #4 for their positive and kind words regarding our manuscript. We believe that our additional work, analysis, and rewording should address many of the concerns.

Comment #4-1: The part “Trem2+Vcam1+ skin cancer-associated myeloid cells (SCAMs) are distinct from other non-cancer-associated Trem2+ myeloid cells” is expected and do not provide real new insights in the study. It is indeed expected to see transcriptomic changes in tumor associated macrophages compared to tissue resident macrophages in steady state or in wounded skin. Also, the use of the name “tricophages” is a bit odd and is not commonly used in the field.

Response #4-1: We thank the reviewer for their comments and insight on the field of macrophage biology. We agree that it is not fully surprising to see differences between SCAMs and other non-cancer-associated Trem2⁺ myeloid cells, however, we feel that we very well define these differences. From our transcriptomic analysis, we identified that SCAMs, as compared to the non-cancer-associated Trem2⁺ myeloid cells, uniquely express the surface marker Vcam1. We additionally found that SCAMs have increased proliferative capacity relative to their non-cancer-associated counterparts. We think it is important to note that the word ‘tricophages’ is not our term but refer the reviewer to the original publications: Wang et al. Cell Stem Cell 2019. These previously described regulatory macrophages have been shown to inhibit hair follicle proliferation in a distinct manner. Overall, we would argue that the identification of unique surface markers and biological associated processes (proliferation) would certainly constitute unique and insightful differences between SCAMs and non-cancer-associated Trem2⁺ cells.

Comment #4-2: The part “SCAMs promote tumor growth through an OSM-OSMR-STAT3 axis” is of interest but not formally validated in vivo using genetic approached removing OSM expression in TREM2⁺ VCAM1⁺ macrophages.

Response #4-2: We thank the reviewer for this comment and are gratified to hear that they find this point interesting. Although it is currently not possible to genetically ablate OSM expression within Trem2⁺Vcam1⁺ cells within our mouse BCC tumor model because we are already using K14-CreER to drive tumor formation within the skin epithelium, we agree that in general, further in vivo validation would enhance this point. As shown above, in **Minor Response #2-5**, we employ an anti-Osm antibody neutralization strategy and found reduced tumor growth, indicating dependence on Osm for tumor growth.

Comment #4-3: The last part is quite confusing. It is not really clear how much cells self-maintain vs are recruited from circulating precursors. More direct quantitative fate mapping model of monocytes could be used to support these conclusions.

Response #4-3: We thank the reviewer for this comment and agree that there is a lot of information presented, with some limitations that we should clarify. Utilization of the CD45.1 and CD45.2 system, we can easily exclude that while tracing the CD45.2⁺, we are only following cells that are self-maintained and not cells that are recruited. Our main goal is to focus on the CD45.2⁺ macrophages rather than to

distinguish between self-maintenance and recruited cells. Our system is not designed to monitor and fate map recruited CD45.1⁺ cells from the NodScid mouse host. Our studies clearly demonstrate that the recruitment of circulating precursors is not necessary for the self-propagation of the allografted CD45.2⁺ cells. We agree that a detailed fate mapping of self-propagation versus recruitment is of great interest, but our current mouse model precludes it. We are in the process of constructing a novel multi-allele, syngeneic model, but these detailed and important kinetic studies will be published in a different setting.

Comment #4-4: Finally, on the use of used RNAScope to look for the male-specific marker Uty to differentiate between transplanted and recipient cells, how the authors excluded phagocytosis of tumor cells?

Response #4-4: We believe that the reviewer is wondering whether it is possible that CD45.1⁺ macrophages from the NodScid host are appearing to be *Uty*⁺ because they are engulfing transplanted male *Uty*⁺ tumor cells. We can't be certain of this, but we feel that our secondary experiment taking advantage of the CD45.1 and CD45.2 system reinforces the same conclusions from the *Uty*-based experiments. We have re-worded our results section however to include the phagocytosis hypothesis before we describe our secondary CD45.1 and CD45.2 based experiments.

Reviewer #5:

In this study, Haensel et al reported how tumors assemble a spatially organized self-propagating Trem2+ myeloid niche. Authors exploited multiple single-cell measurements including CODEX, scRNA-seq, and CYTOF in primary human and mouse tissues to better define microenvironmental neighborhoods surrounding BCC tumor epithelia. They found that TREM+ skin cancer related macrophages enable proliferation of a specific tumor epithelia population via JAK/STAT signaling. Using mouse models to test hypotheses generated from data in human tumors, they reported that tumors drives the differentiation of skin cancer related macrophages. This work took advantage of cutting-edge single-cell strategies and relied on mouse models to fully assess the relevance of Trem2+ myeloid niche in tumors. Single-cell RNA-seq data in different mouse models were generated to assess the relevance of Trem+ cells in tumor growth.

We thank the reviewer for their positive support of the work.

Comment #5-1: Authors can improve the quality of their study by comparing the scRNA-seq data generated throughout their study to find similarities and differences across different in vivo models. Moreover, gene lists associated with different experimental systems can enable the community to better utilize the data generated in this study.

Response #5-1: We thank the reviewer for this suggestion. We have gone ahead and expanded our analysis from Fig. 3 of this paper where we compared myeloid cells from normal skin, wounded skin, and our BCCs to include additional datasets. We have included a few other well-known datasets in the field of Trem2 biology including microglia, cells from a sarcoma model, and cells from a colon cancer model. We first started by simply subsetting the different myeloid fractions from these models before subsetting on Trem2-expressing cells. We found that in general, the different *Trem2*⁺ cells from the cancer models clustered near one another, the normal and wounded skin cells clustered near one another, and the microglia clustered away from all the samples. Differential gene expression found that each sample had distinct marker genes. Furthermore, we have included the gene lists in our revised manuscript. We have added this new data found in Reviewer Figure 9 to Extended Data Figure 4.

Reviewer Figure 9: (A) Overview and color labels for the different merged datasets. (B) UMAP of the merged myeloid fractions from each of the datasets. (C) Feature plot for *Trem2*. (D) Subset UMAP of *Trem2*-expressing cells from (B). (E) Feature plot for *Trem2* of the subset. (F) Heatmap showing the top marker genes of the various *Trem2*-expressing cells from the different datasets.

REVIEWERS' COMMENTS

Reviewer #1 (Remarks to the Author):

I am satisfied with the responses provided by the authors to my previous concerns

Reviewer #2 (Remarks to the Author):

I would like to congratulate the authors and thank them for the additional experiments performed to address remaining concerns.

Although the consequences of anti-TREM2 depletion cannot be fully addressed due to the models available, I believe this work provides substantial advance in the field to be accepted.

Reviewer #3 (Remarks to the Author):

No text inputted here. Please see comments to editors.

Reviewer #4 (Remarks to the Author):

The authors have addressed all the concerns that were raised.